# Ribosome binding protein GCN1 regulates the cell cycle and cell proliferation and is essential for the embryonic development of mice

Hiromi Yamazaki[1], Shuya Kasai[1], Junsei Mimura[1], Peng Ye[1], Atsushi Inose-Maruyama[1¤], Kunikazu Tanji[2], Koichi Wakabayashi[2], Seiya Mizuno[3], Fumihiro Sugiyama[3], Satoru Takahashi[3], Tsubasa Sato[1,4], Taku Ozaki[4], Douglas R. Cavener[5], Masayuki Yamamoto[6], Ken Itoh[1]*

1 Department of Stress Response Science, Center for Advanced Medical Research, Hirosaki University, Hirosaki, Japan, 2 Department of Neuropathology, Institute of Brain Science Graduate School of Medicine, Hirosaki University, Hirosaki, Japan, 3 Transborder Medical Research Center and Laboratory Animal Resource Center, University of Tsukuba, Tsukuba, Japan, 4 Department of Chemistry and Biological Sciences, Faculty of Science and Engineering, Iwate University, Morioka, Japan, 5 Department of Biology, Center for Cellular Dynamics and the Huck Institute of the Life Sciences, Pennsylvania State University, University Park, Pennsylvania, United States of America, 6 Department of Medical Biochemistry, Tohoku University Graduate School of Medicine, Sendai, Japan

¤ Current address: Division of Microbiology and Molecular Cell Biology, Nihon Pharmaceutical University
* itohk@hirosaki-u.ac.jp

**Data Availability Statement:** All relevant data are within the manuscript and its Supporting Information files.

## Abstract

Amino acids exert many biological functions, serving as allosteric regulators and neurotransmitters, as constituents in proteins and as nutrients. GCN2-mediated phosphorylation of eukaryotic initiation factor 2 alpha (eIF2α) restores homeostasis in response to amino acid starvation (AAS) through the inhibition of the general translation and upregulation of amino acid biosynthetic enzymes and transporters by activating the translation of Gcn4 and ATF4 in yeast and mammals, respectively. GCN1 is a GCN2-binding protein that possesses an RWD binding domain (RWDBD) in its C-terminus. In yeast, Gcn1 is essential for Gcn2 activation by AAS; however, the roles of GCN1 in mammals need to be established. Here, we revealed a novel role of GCN1 that does not depend on AAS by generating two *Gcn1* mutant mouse lines: *Gcn1*-knockout mice (*Gcn1* KO mice (*Gcn1$^{-/-}$*)) and RWDBD-deleted mutant mice *(Gcn1$^{\Delta RWDBD}$ mice)*. Both mutant mice showed growth retardation, which was not observed in the *Gcn2* KO mice, such that the *Gcn1* KO mice died at the intermediate stage of embryonic development because of severe growth retardation, while the *Gcn1$^{\Delta RWDBD}$* embryos showed mild growth retardation and died soon after birth, most likely due to respiratory failure. Extension of pregnancy by 24 h through the administration of progesterone to the pregnant mothers rescued the expression of differentiation markers in the lungs and prevented lethality of the *Gcn1$^{\Delta RWDBD}$* pups, indicating that perinatal lethality of the *Gcn1$^{\Delta RWDBD}$* embryos was due to simple growth retardation. Similar to the yeast Gcn2/Gcn1 system, AAS- or UV irradiation-induced eIF2α phosphorylation was diminished in the *Gcn1$^{\Delta RWDBD}$* mouse embryonic fibroblasts (MEFs), suggesting that GCN1 RWDBD is responsible for GCN2 activity. In addition, we found reduced cell proliferation and G2/M arrest accompanying a decrease in Cdk1 and Cyclin B1 in the *Gcn1$^{\Delta RWDBD}$* MEFs. Our

**Funding:** This work was supported by grants from the Japan Society for the Promotion of Science (JSPS) (KAKENHI 26111010 and 25293064 to K.I. and 26860178 and 17K08616 to H.Y.), funds from the Naito Foundation and the Karoji Memorial Fund for Medical Research. The funders had no role in study design, data collection and analysis, decision to publish, or preparation of the manuscript.

**Competing interests:** The authors have declared that no competing interests exist.

results demonstrated, for the first time, that GCN1 is essential for both GCN2-dependent stress response and GCN2-independent cell cycle regulation.

## Author summary

The stress response at the translational level is an energetically cost-saving mechanism because translation consumes a considerable amount of energy. Upon exposure to stresses such as that from amino acid starvation (AAS), the translational initiation factor eIF2α is phosphorylated, which represses general translation to save energy. At the same time, eIF2α phosphorylation increases the selective translation of cytoprotective proteins, such as ATF4, that transcriptionally activate the stress response, promoting cell survival. Among four eIF2α kinases, GCN2 responds to AAS and phosphorylates eIF2α. In yeast, Gcn1 is required for Gcn2 activation by AAS, but the roles of GCN1 in mammals remain to be established. Here, we show that GCN1 is involved in GCN2-mediated eIF2α phosphorylation after AAS and UV radiation by generating *Gcn1* mutant mice. Interestingly, GCN1 not only regulates the eIF2α-mediated stress response but also the cell cycle and cell proliferation in a GCN2-independent manner. Taking these findings together, we propose that GCN1 integrates cellular information and coordinates the cellular stress response to enhance viability.

## Introduction

Translational regulation through the phosphorylation of eukaryotic initiation factor 2 alpha (eIF2α) at Ser51 is instigated by a wide range of stresses that regulate protein synthesis and generate cytoprotective responses and therefore is called the integrated stress response (ISR) [1,2]. In mammalian cells, four eIF2α kinases, GCN2, PERK, PKR, and HRI, have been identified, and they are activated by various stresses, such as amino acid starvation (AAS), ER stress, virus infection and heme deficiency, respectively [1,2]. Under stressed conditions, phosphorylated eIF2α reduces the amount of GTP-bound eIF2 by competitively inhibiting eIF2B and repressing general translation. On the other hand, eIF2α phosphorylation increases the translation of a range of mRNAs having an upstream open reading frame in the 5'-UTR, such as the transcription factor *ATF4* [1,2]. ATF4 regulates various target genes related to amino acid synthesis, amino acid transport, apoptosis and autophagy to maintain amino acid homeostasis [3, 4]. GCN2 might be the most ancient eIF2α kinase in eukaryotes found in yeasts, plants and mammals, and budding yeast *Saccharomyces cerevisiae* (*S. cerevisiae*) possess Gcn2 as the sole eIF2α kinase [5]. The function of Gcn1 in response to AAS has been extensively studied in yeasts [6–8]. Upon AAS, uncharged tRNA binds to the HisRS-like domain located at the C-terminus of Gcn2 and leads to its activation. Gcn1 bound to the ribosome presumably transfers the uncharged tRNA at the A-site of the ribosome to Gcn2 and thus is essential for Gcn2 activation by AAS [7].

In *S. cerevisiae*, divergent stresses other than AAS such as glucose deprivation and high salt activates Gcn4 in Gcn2-tRNA binding motif dependent manner indicating the accumulation of uncharged tRNA as the responsible mechanism [1]. Also in *Schizosaccharomyces pombe (S. Pombe)*, uncharged tRNA binding to Gcn2 is essential for the response to AAS as well as to hydrogen peroxide (H₂O₂) and UV radiation, and these responses require Gcn1 [9]. Recent analysis also showed that GCN2 mediates ISR activation by the unfolded protein response in

mitochondria (UPR$^{mt}$) in *Caenorhabditis elegans* (*C. elegans*) [10] and that its activation leads to longevity expansion in both *C. elegans* and *Drosophila* [10,11]. In mammals, GCN2 was shown to be essential for AAS-mediated ATF4 activation. An initial study of *Gcn2* KO mice against a heterozygous mouse background reported that these mutants are viable under normal laboratory conditions [12], but many died shortly (1–2 days) after birth by unknown mechanisms in C57BL/6J background [13]. They also showed increased mortality [13] and exaggerated oxidative stress under AAS [14]. GCN2 regulates a wide variety of physiological responses, including those involved in feeding behavior, memory formation, fatty acid metabolism and inflammation [15], and phosphorylates various target proteins in addition to eIF2α [16,17]. In mammals, GCN2 is also activated by mitochondrial stress [18] or glucose starvation [19], and in human, *GCN2* mutation is related to pulmonary hypertension [20]. Although most of the stress-induced GCN2 activation in multicellular eukaryotes may be mediated by AAS or an increase in uncharged tRNA, similar to the action in yeast, it has been proposed that GCN2 may require an additional independent signal other than AAS in T cells [21]. The activation mechanism of GCN2 in the above-mentioned circumstances remains to be determined.

Mammalian GCN1 is essential for GCN2 activation by AAS and other GCN2-mediated stresses, including UV exposure [22]. However, the roles of GCN1 in mammalian physiology have been rarely explored. Gcn1 binds to several proteins, including Gcn2, Gir2 (its mammalian homolog is DFRP2), and Yih1 (its mammalian homolog is IMPACT) through its RWD binding domain (RWDBD) [1,23] (see Fig 1B). Recent studies in various species have also reported GCN2-independent roles of GCN1. GCN1 regulates apoptosis in *C. elegans* [24] and regulates the innate immune response in *Arabidopsis thaliana* (*A. thaliana*) in a GCN2-independent but GCN20-dependent manner [25]. To clarify the role of GCN1 in mammals, we created two *Gcn1* mutant mouse lines, one with exon 2 of the *Gcn1* gene deleted (*Gcn1* KO mice (*Gcn1$^{-/-}$*)) and the other with exons deleted between 46 and 53, which corresponds to RWDBD of GCN1 (*Gcn1$^{ΔRWDBD/ΔRWDBD}$* mice). Interestingly, these two *Gcn1* mutant mice showed growth retardation and lethality, which was not observed for the *Gcn2* KO mice, suggesting that GCN1 may have additional roles other than the AAS response in mammals.

## Results

### *Gcn1$^{ΔRWDBD}$* mice exhibited growth retardation and died perinatally

To generate *Gcn1* KO mice (*Gcn1$^{-/-}$*), we first deleted exon 2 of the *Gcn1* gene by using the CRISPR/Cas9 system (Fig 1A and 1B). In contrast to *Gcn2* KO mice, we found that *Gcn1$^{-/-}$* mice experience embryonic lethality since no *Gcn1$^{-/-}$* embryos were found around embryonic days E15.5 to E16.5 in pregnant *Gcn1$^{+/-}$* females crossed with *Gcn1$^{+/-}$* males (Table 1). *Gcn1* KO embryos could be found at E10.5, but they were extremely small compared to the WT embryos, which may have been caused by severe growth retardation (Fig 1C). In addition, we observed that *Gcn1* mRNA was expressed in embryos from E9.5 to E14.5 in all the organs that we examined (S1 Fig). These observations suggest that GCN1 plays a specific role in embryonic development beyond that of GCN2. Therefore, we next generated another *Gcn1* mutant mouse specifically lacking RWDBD, which is important for GCN2 binding, by deleting the exons between 46 and 53 by the CRISPR/Cas9 system (Fig 1D and 1E). We verified the deletion by PCR coupled with sequence analysis of the PCR-amplified genomic DNA and obtained a total of 127 pups carrying the *Gcn1$^{ΔRWDBD}$* allele. These founder mice were mated with wild-type (WT) mice, and 4 independent lines were established. *Gcn1$^{+/ΔRWDBD}$* mice were viable and fertile and we intercrossed these mice to obtain 80 mice that survived for at least 2 weeks. Surprisingly, no *Gcn1$^{ΔRWDBD/ΔRWDBD}$* mice (hereafter, called *Gcn1$^{ΔRWDBD}$* mice) survived to this stage (Table 2). To determine the time of lethality, we analyzed the embryos at different

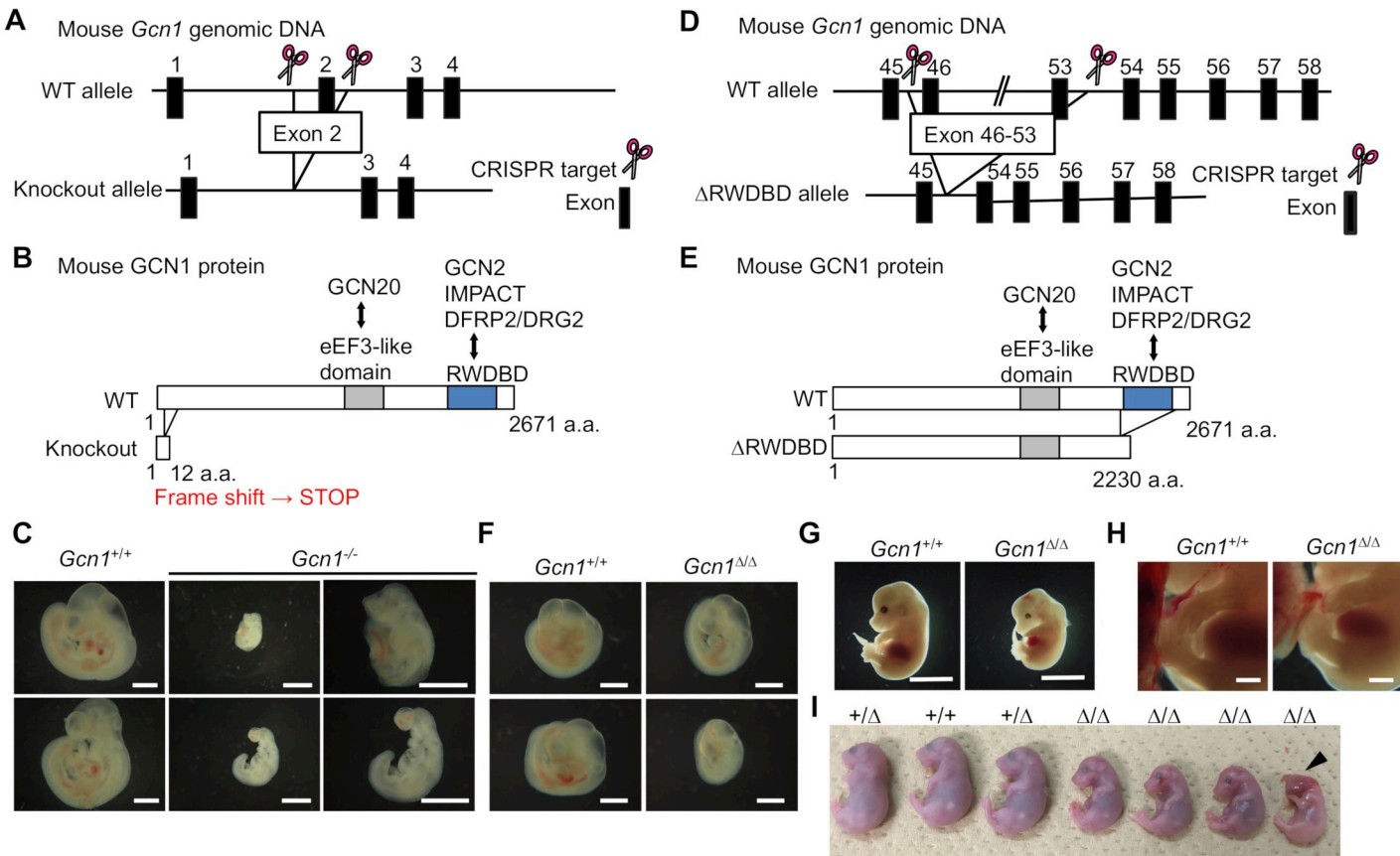

**Fig 1. Establishment of *Gcn1* KO and *Gcn1*^*ΔRWDBD*^ mice.** (A) Schematic structure of the *Gcn1* KO allele. *Gcn1* KO mice were generated by the CRISPR/Cas9 system. Double-strand breaks were induced within introns 1 and 2, and exon 2 was excised. (B) Schematic structure of the mutant GCN1 protein produced in *Gcn1* KO mice. In *Gcn1* KO mice, exon 2 of the *Gcn1* gene was deleted, resulting in a frameshift and premature stop codon in exon 3. Thus, *Gcn1* KO mice only expressed a short form of the GCN1 protein, consisting of the N-terminus 12 amino acids (a.a.) and lacked a well-conserved eEF3-like domain and RWD binding domain. (C) Representative pictures of the *Gcn1* KO embryos at E10.5. Enlarged pictures of the *Gcn1* KO embryos are shown in the right panel. Scale bar: 1 mm. (D) Schematic structure of the *Gcn1*^*ΔRWDBD*^ allele. *Gcn1*^*ΔRWDBD*^ mice were generated by the CRISPR/Cas9 system. Double-strand breaks were induced within introns 45 and 53, and exons 46–53 were excised. (E) Schematic structure of the mutant GCN1 protein produced in *Gcn1*^*ΔRWDBD*^ mice. GCN1^*ΔRWDBD*^ mice lack the RWD binding domain. (F) Representative pictures of the embryos at E9.5. Scale bar: 1 mm. (G) Representative pictures of the embryos at E14.5. Scale bar: 5 mm. (H) Enlarged pictures of the embryos at E14.5. Limb development was delayed in the *Gcn1*^*ΔRWDBD*^ embryo. Scale bar: 5 mm. (I) Representative pictures of the embryos at E18.5. The arrowhead indicates the abnormality of the head or an anencephaly-like phenotype.

developmental stages after the timed mating of the *Gcn1*^*+/ΔRWDBD*^ mice. In contrast to the *Gcn1*^*-/-*^ embryos, the *Gcn1*^*ΔRWDBD*^ embryos were found during every stage of embryonic development (Table 3). However, the *Gcn1*^*ΔRWDBD*^ embryos were smaller than the embryos of

**Table 1. Genotypes of viable offspring from *Gcn1*^*+/-*^ mouse intercrosses.**

| Stage | Parameter | Number of offspring by genotype | | | Total |
|---|---|---|---|---|---|
| | | *Gcn1*^*+/+*^ | *Gcn1*^*+/−*^ | *Gcn1*^*−/−*^ | |
| E10.5 | Predicted | 7 | 13 | 7 | 26 |
| | Observed | 7 | 14 | 5 | |
| E15.5 | Predicted | 4 | 8 | 4 | 15 |
| | Observed | 5 | 10 | 0 | |
| E16.5 | Predicted | 2 | 4 | 2 | 7 |
| | Observed | 3 | 4 | 0 | |

**Table 2. Genotypes of viable offspring from *Gcn1*<sup>+/ΔRWDBD</sup> mouse intercrosses.**

| Parameter at 2 weeks | Number of offspring of genotype | | | |
|---|---|---|---|---|
| | *Gcn1*$^{+/+}$ | *Gcn1*$^{+/\Delta}$ | *Gcn1*$^{\Delta/\Delta}$ | Total |
| Predicted | 32 | 64 | 32 | 128 |
| Observed | 47 | 80 | 0 | 127 |

*Gcn1*$^{+/\Delta RWDBD}$ and their *Gcn1*$^{+/+}$ littermates between E9.5 and E18.5 (Fig 1F, 1G and 1I) and showed a statistically significant decrease in body weight after E17.5 (Fig 2B). Although growth retardation was observed, the *Gcn1*$^{\Delta RWDBD}$ embryos were apparently normal, although some of these embryos either showed abnormalities in the head, an anencephaly-like phenotype at E14.5 (S2A Fig) or E18.5 (Fig 1I). They also showed a developmental delay in the limbs at E14.5, which was not observed in the *Gcn1*$^{+/+}$ embryos (Fig 1H). To determine the cause of lethality, we then examined the *Gcn1*$^{\Delta RWDBD}$ embryos at E18.5 (i.e., one day before birth) after cesarean section. Of note, *Gcn1*$^{\Delta RWDBD}$ mice at E18.5 did not respire and died within 20 min after delivery by cesarean section, while almost all *Gcn1*$^{+/+}$ and *Gcn1*$^{+/\Delta RWDBD}$ embryos respired and their color turned red. We next tried to prolong the gestational period of pregnant mothers by progesterone administration because progesterone maintains pregnancy (Fig 2A). The *Gcn1*$^{\Delta RWDBD}$ embryos at E19.5 and E20.5 still showed lower body weight compared to the *Gcn1*$^{+/+}$ and *Gcn1*$^{+/\Delta RWDBD}$ embryos, although they gradually increased their body weight (Fig 2B). At E20.5, we observed 5 *Gcn1*$^{\Delta RWDBD}$ embryos with normal appearance and 4 *Gcn1*$^{\Delta RWDBD}$ embryos with mandibular hypoplasia and exophthalmos/hypoplasia of the eyelid (Fig 2C and S2B and S2C Fig). The *Gcn1*$^{\Delta RWDBD}$ embryos with abnormal appearance died within 20 min after delivery by cesarean section, whereas 2 of 5 *Gcn1*$^{\Delta RWDBD}$ embryos with normal appearance respired, turned red, and survived for at least 4 h (Fig 2C). To further examine the cause of the respiratory failure, we analyzed lung development in the *Gcn1*$^{\Delta RWDBD}$ embryos at E18.5 by a lung float test and histological analysis. Curiously, the lungs from the

**Table 3. Genotypes of viable offspring from *Gcn1*<sup>+/ΔRWDBD</sup> mouse intercrosses.**

| Stage | Parameter | Number of offspring by genotype | | | Total |
|---|---|---|---|---|---|
| | | *Gcn1*$^{+/+}$ | *Gcn1*$^{+/\Delta}$ | *Gcn1*$^{\Delta/\Delta}$ | |
| E9.5 | Predicted | 2 | 4 | 2 | 8 |
| | Observed | 1 | 4 | 3 | |
| E11.5 | Predicted | 4 | 9 | 4 | 17 |
| | Observed | 6 | 8 | 3 | |
| E14.5 | Predicted | 7 | 14 | 7 | 27 |
| | Observed | 7 | 16 | 4 | |
| E16.5 | Predicted | 2 | 3 | 2 | 7 |
| | Observed | 3 | 2 | 1 | |
| E17.5 | Predicted | 2 | 5 | 2 | 9 |
| | Observed | 2 | 3 | 4 | |
| E18.5 | Predicted | 12 | 24 | 12 | 47 |
| | Observed | 12 | 28 | 7 | |
| E19.5 | Predicted | 5 | 10 | 5 | 19 |
| | Observed | 7 | 8 | 4 | |
| E20.5 | Predicted | 10 | 19 | 10 | 38 |
| | Observed | 4 | 27 | 9 | |

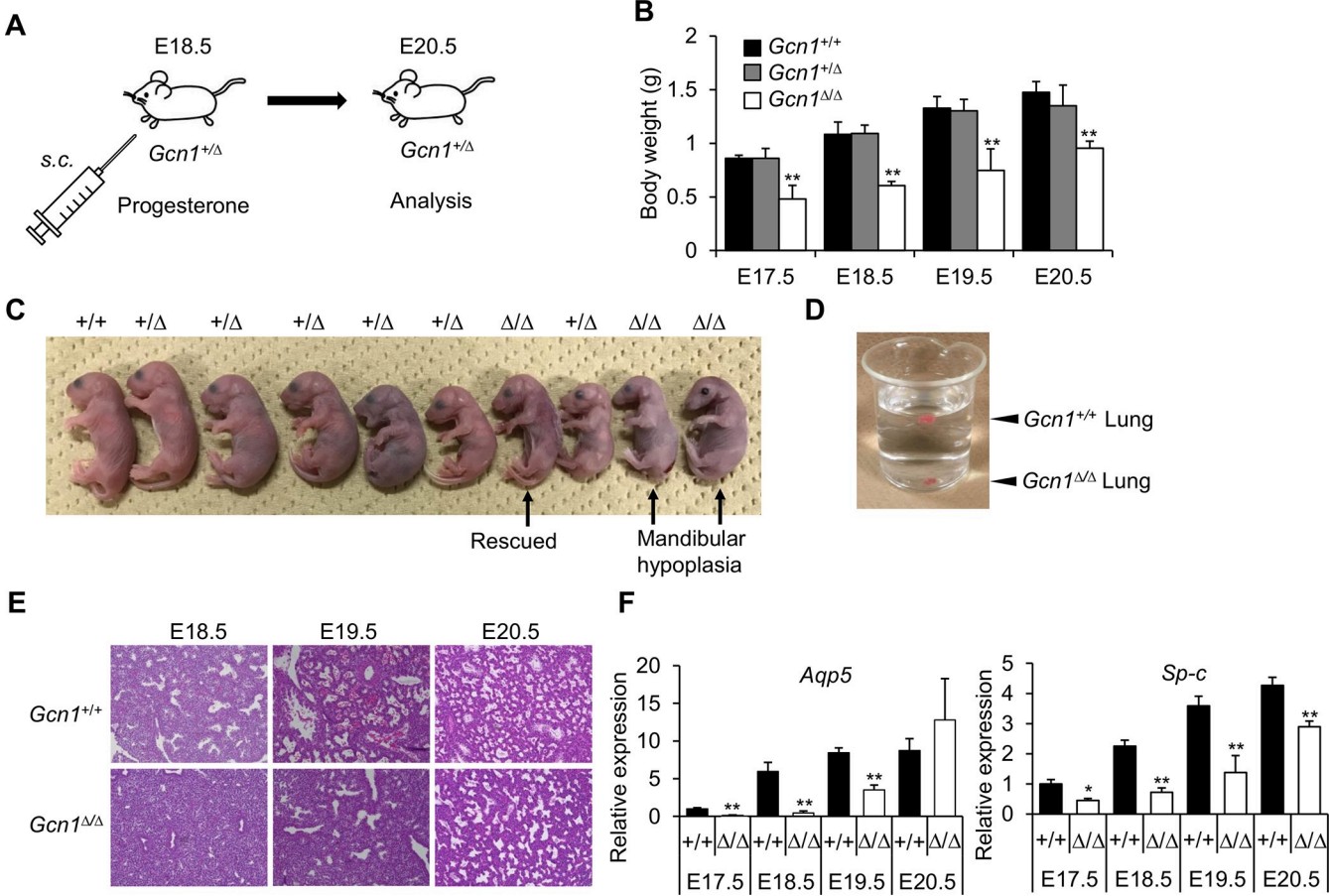

**Fig 2. Gcn1$^{ΔRWDBD}$ mice show severe growth retardation and perinatal lethality.** (A) Procedure for prolongation of gestation period. (B) Body weight for each embryo at the indicated stages. The data of E19.5 and E20.5 embryos were from those mice in which the gestation period was prolonged. Data are shown as means±SD from multiple independent animals. WT: E17. 5 N = 2, E18.5 N = 11, E19.5 N = 7, E20.5 N = 4. Gcn1$^{+/ΔRWDBD}$: E17.5 N = 3, E18.5 N = 3, E19.5 N = 8, E20.5 N = 18. Gcn1$^{ΔRWDBD/ΔRWDBD}$: E17.5 N = 4, E18.5 N = 23, E19.5 N = 4, E20.5 N = 5. * $p<0.05$, ** $p<0.01$ compared with the WT (two tailed Student's $t$-test). (C) Representative pictures of the embryos at E20.5. (D) Beaker containing excised lungs in PBS from E18.5 embryos. Gcn1$^{ΔRWDBD}$ lungs sunk to the bottom of the container, whereas WT (Gcn1$^{+/+}$) lungs floated at the PBS surface. (E) Hematoxylin-eosin (HE) staining of the lungs of Gcn1$^{+/+}$ and Gcn1$^{ΔRWDBD}$ at the indicated stages. At least 3 independent animals were subjected to the analysis and representative data was shown. The data of E19.5 and E20.5 embryos came from in which the gestation period was prolonged. (F) The expression of Aqp5 (alveolar type I (AT1) marker) and Sp-c (type II (AT2) makers) in the lungs of WT (Gcn1$^{+/+}$) and Gcn1$^{ΔRWDBD}$ were quantified by RT-PCR. The data of E19.5 and E20.5 embryos came from in which the gestation period was prolonged. Data are shown as means±SD from multiple independent animals (WT: E17. 5 N = 2, E18.5 N = 6, E19.5 N = 4, E20.5 N = 4. Gcn1$^{ΔRWDBD}$: E17.5 N = 3, E18.5 N = 3, E19.5 N = 3, E20.5 N = 4). * $p<0.05$, ** $p<0.01$ compared with the WT (two tailed Student's $t$-test).

Gcn1$^{+/+}$ embryos could float on phosphate-buffered saline (PBS), but those from the Gcn1$^{ΔRWDBD}$ embryos sunk to the bottom of the container holding PBS (Fig 2D). Histological analysis also revealed alveolar collapse of the Gcn1$^{ΔRWDBD}$ lungs (Fig 2E), suggesting that the lungs from the Gcn1$^{ΔRWDBD}$ embryos lacked aeration. We then analyzed lung differentiation markers and found that the Gcn1$^{ΔRWDBD}$ lungs showed lower expression of Aqp5 and Sp-c genes compared to that in the Gcn1$^{+/+}$ lungs on the same embryonic day (Fig 2F). In addition, lungs from the Gcn1$^{ΔRWDBD}$ mice at E20.5 appeared similar to those of the control mice, and the gene expression level of alveolar markers in the Gcn1$^{ΔRWDBD}$ mice at E20.5 was comparable to those of the WT embryos at E18.5 (Fig 2E and 2F). Collectively, these results demonstrated that the delay in lung development was most likely the cause of lethality of the Gcn1$^{ΔRWDBD}$ embryos, although an overall developmental delay was also observed.

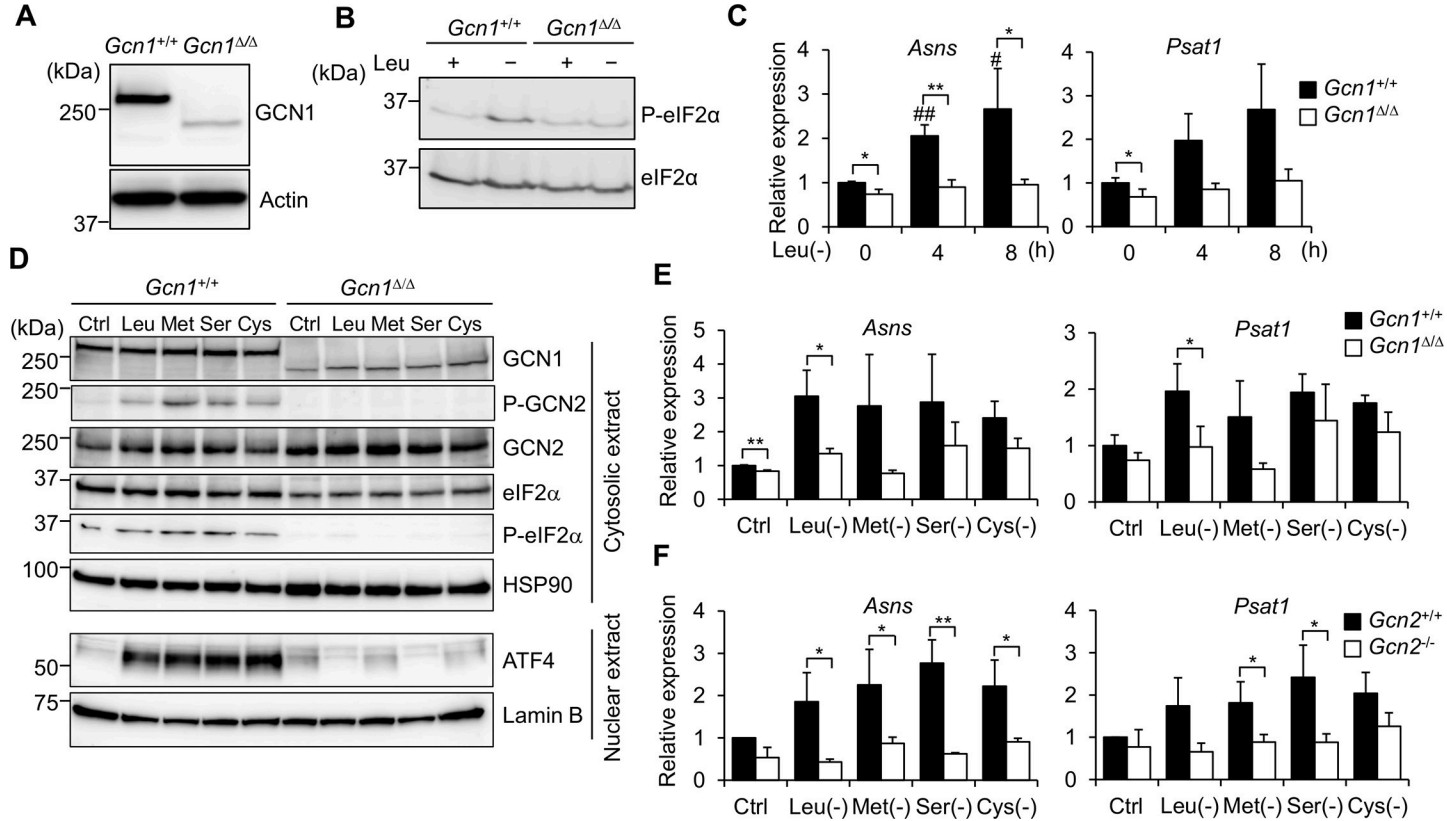

**Fig 3. GCN1 is necessary for GCN2-mediated ATF4 activation.** (A) Whole cell protein extracted from the *Gcn1*^+/+^ and *Gcn1*^ΔRWDBD^ MEFs was subjected to immunoblot analysis to detect GCN1. (B) The WT (*Gcn1*^+/+^) and *Gcn1*^ΔRWDBD^ MEFs were exposed to leucine starvation for 4 h, and the phosphorylation level of eIF2α was examined by immunoblot. (C) The *Gcn1*^+/+^ and *Gcn1*^ΔRWDBD^ MEFs were exposed to leucine starvation for the indicated times, and the mRNA levels of the ATF4 target genes *Asns* and *Psat1* were quantified by RT-PCR. The value at 0 h for WT cells was set to 1, and the results were shown as the relative folds±SD from multiple independent experiments (N = 4). * $p < 0.05$, ** $p < 0.01$ compared with the WT (two tailed Student's *t*-test). # $p < 0.05$, ## $p < 0.01$ compared with the value at 0 h for the WT cells (one-way ANOVA with with Bonferroni post hoc test). (D) The WT (*Gcn1*^+/+^) and *Gcn1*^ΔRWDBD^ MEFs were exposed to leucine (Leu), methionine (Met), serine (Ser) or cystine (Cys) starvation for 4 h or cultured in the control (Ctrl) medium and cells were fractionated into cytosol, nuclear fractions and subjected to immunoblot analysis to detect the GCN1, phosphorylated GCN2 (P-GCN2), GCN2, phosphorylated eIF2α (P-eIF2α), eIF2α, HSP90, ATF4 and Lamin B. (E) The WT (*Gcn1*^+/+^) and *Gcn1*^ΔRWDBD^ MEFs lacking the indicated amino acids were exposed to AAS for 8 h, and the mRNA levels of *Asns* and *Psat1* were quantified by RT-PCR. The value for WT cells cultured in the control (Ctrl) medium was set to 1, and the results were shown as the relative folds±SD from multiple independent experiments (N = 4). * $p < 0.05$, ** $p < 0.01$ compared with the WT (two tailed Student's *t*-test). (F) The WT (*Gcn2*^+/+^) and *Gcn2* KO (*Gcn2*^-/-^) MEFs were exposed to AAS lacking the indicated amino acids for 8 h, and the mRNA levels of *Asns* and *Psat1* were quantified by RT-PCR. The value for WT cells cultured in the control (Ctrl) medium was set to 1, and the results were shown as the relative folds±SD from multiple independent experiments (N = 4). * $p < 0.05$, ** $p < 0.01$ compared with the WT (two tailed Student's *t*-test).

## Expression of GCN1 in WT and *Gcn1*^ΔRWDBD^ MEFs

To investigate the effect of RWDBD deletion of GCN1 on cellular function, we established mouse embryonic fibroblasts (MEFs) from the *Gcn1*^ΔRWDBD^ embryos. As expected, the *Gcn1*^ΔRWDBD^ MEFs expressed a truncated type of GCN1 protein, but the level of the ΔRWDBD GCN1 protein decreased to approximately 30% compared to that of the WT MEFs (Fig 3A). As previous results indicated a role for GCN1 in the nucleus [26], we examined the subcellular localization of GCN1. Immunocytochemical analysis in HeLa cells showed that GCN1 mainly localized to the cytoplasm (S3A Fig); however, GCN1 exactly colocalized neither with the ER marker calnexin nor with the mitochondrial marker pyruvate dehydrogenase (S3B and S3C Fig). Cell extract fractionation analysis also showed that GCN1 localized to the cytoplasm and rarely localized to the nucleus in HeLa cells (S3D Fig), and both in WT and *Gcn1*^ΔRWDBD^ MEFs (S3E Fig).

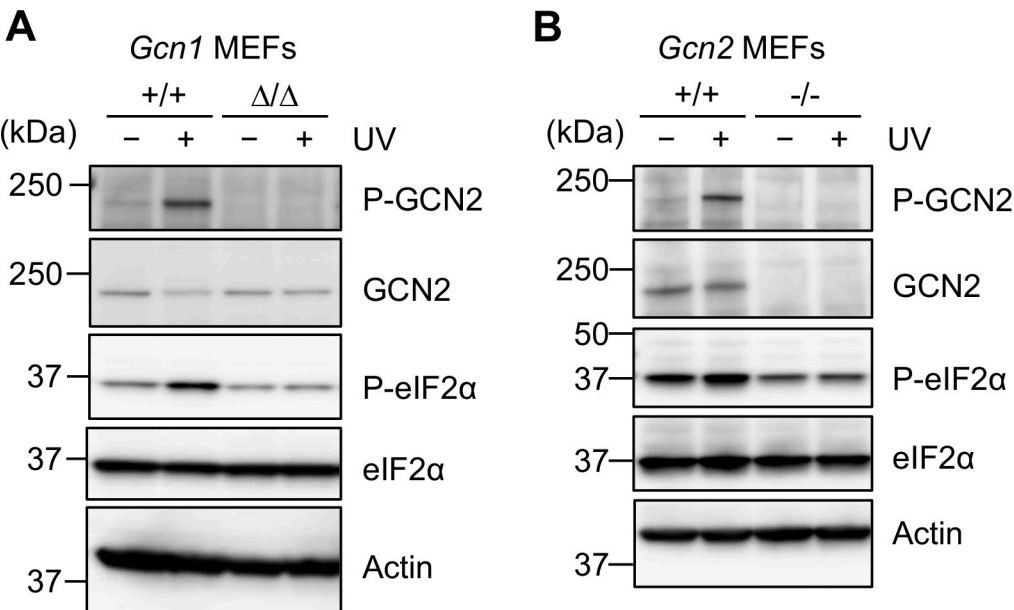

**Fig 4. GCN1 and GCN2 dependency in response to UV exposure.** (A)(B) The WT and $Gcn1^{\Delta RWDBD}$ (A) or $Gcn2$ KO ($Gcn2^{-/-}$) MEFs (B) were irradiated by 80 J/m$^2$ UV and were allowed to recover in fresh medium for 30 min or 4 h before harvest. Whole cell protein extracts of the MEFs were subjected to immunoblot analysis to detect phosphorylated GCN2 (P-GCN2), GCN2, phosphorylated eIF2α (P-eIF2α), eIF2α and β-actin.

## GCN1 was necessary for the AAS response of MEFs

Next, we examined the role of GCN1 in the GCN2-eIF2α pathway in the MEFs. Under basal conditions, the global translation level in the $Gcn1^{\Delta RWDBD}$ MEFs was comparable to that in the WT MEFs (S4 Fig). Then, we analyzed the role of GCN1 in response to AAS. eIF2α phosphorylation was increased in the WT MEFs but was unchanged in the $Gcn1^{\Delta RWDBD}$ MEFs (Fig 3B and quantified in S5A Fig). Upon leucine starvation, the expression of ATF4 target genes asparagine synthetase (*Asns*) and phosphoserine aminotransferase 1 (*Psat1*) was activated by leucine starvation in the WT MEFs, while the induction of these genes was suppressed in the $Gcn1^{\Delta RWDBD}$ MEFs (Fig 3C). We further explored the role of GCN1 in activating the GCN2/ATF4 pathway by depleting amino acids other than leucine. Phosphorylation of GCN2 and eIF2α and nuclear accumulation of ATF4 were induced by depletion of leucine, methionine, serine or cystine in WT MEFs, which was not observed in $Gcn1^{\Delta RWDBD}$ MEFs (Fig 3D and S5B Fig). Both *Asns* and *Psat1* genes were induced by depletion of methionine, serine or cystine in the WT MEFs, whereas the induction was largely suppressed in both $Gcn1^{\Delta RWDBD}$ and $Gcn2$ KO MEFs (Fig 3E and 3F). These data suggested that GCN1 is necessary for GCN2-dependent ATF4 activation upon AAS in mammalian cells.

## GCN1 was also necessary for the GCN2-mediated response to UV stress

UV radiation increases eIF2α phosphorylation in a GCN2-dependent manner [27]. Therefore, we next examined the role of GCN1 in response to UV stress. In the WT MEFs, GCN2 and eIF2α were phosphorylated after 30 min of UV irradiation as previously reported (Fig 4A and 4B and quantified in S6 Fig). On the other hand, inducible phosphorylation of GCN2 and eIF2α was impaired in both $Gcn1^{\Delta RWDBD}$ MEFs and $Gcn2$ KO MEFs (Fig 4A and 4B and quantified in S6 Fig), indicating that GCN1 is also involved in the UV stress response.

## The role of GCN1 in eIF2α phosphorylation by HRI, PERK and PKR

We further explored other stimulations that activate other eIF2α kinases, HRI, PERK and PKR. It has been reported that $H_2O_2$ and the proteasome inhibitor MG132 activate HRI and phosphorylate eIF2α in MEFs [28, 29]. $H_2O_2$ treatment induced phosphorylation of eIF2α in WT, *Gcn1$^{ΔRWDBD}$* and *Gcn2* KO MEFs (S7A–S7B Fig). It is well known that the ER stressor tunicamycin (Tm) induces eIF2α phosphorylation via PERK activation [30]. Tm induced eIF2α phosphorylation in WT, *Gcn1$^{ΔRWDBD}$* and *Gcn2* KO MEFs (S7C and S7D Fig). We further analyzed ATF4 target genes, *Asns* and *Psat1* upon Tm treatment (S6E Fig). Tm-induced *Asns* and *Psat1* mRNA levels in *Gcn1$^{ΔRWDBD}$* MEFs were comparable to those in WT MEFs. We then examined the role of GCN1 in PKR activation. dsRNA mimic Poly(I:C) induced eIF2α phosphorylation in the absence of GCN1 and GCN2 (S7F and S7G Fig). These results indicate that GCN1 is dispensable for HRI, PERK and PKR activation.

## *Gcn1$^{ΔRWDBD}$* MEFs showed decreased growth rate and arrested in G2/M

As the *Gcn1$^{ΔRWDBD}$* embryos exhibited growth retardation (Figs 1 and 2), we examined the cell growth capacity of the *Gcn1$^{ΔRWDBD}$* MEFs. As expected, the cell growth of the *Gcn1$^{ΔRWDBD}$* MEFs was significantly reduced compared to that of the WT MEFs in both primary (Fig 5A) and immortalized MEFs (Fig 5B). Consistent with the decreased cell growth in the *Gcn1$^{ΔRWDBD}$* MEFs, BrdU incorporation was substantially reduced in the *Gcn1$^{ΔRWDBD}$* MEFs compared to the BrdU incorporation in the WT MEFs (Fig 5C). We also analyzed the levels of cleaved PARP and Caspase-3, both of which indicate apoptosis, in the *Gcn1$^{ΔRWDBD}$* MEFs (S8A–S8C Fig). The levels of cleaved PARP and Caspase-3 both in the primary and immortalized *Gcn1$^{ΔRWDBD}$* MEFs were comparable to those of the WT MEFs (S8A–S8C Fig). Flow cytometric analysis revealed that the *Gcn1$^{ΔRWDBD}$* MEFs were significantly larger in size compared to the WT MEFs (Fig 5D). Cell enlargement is often associated with cellular senescence [31]; however, β-galactosidase staining showed similar intensities in both WT and *Gcn1$^{ΔRWDBD}$* MEFs (S9 Fig). Next, we performed cell cycle analysis of the *Gcn1$^{ΔRWDBD}$* MEFs by staining them with propidium iodide (PI) and analyzing them with flow cytometry. The proportion of cells in G2/M phase was increased in the *Gcn1$^{ΔRWDBD}$* MEFs compared to that in the WT MEFs (Fig 5E and 5F). On the other hand, the rate of cell growth and the cell cycle profile were comparable between the *Gcn2* KO MEFs and the WT MEFs (Fig 5G–5I).

## *Cdk1* decreased and *p21* increased in *Gcn1$^{ΔRWDBD}$* MEFs

The RWD domain-containing protein, DRG family-regulatory protein 2 (DFRP2), is a human homolog of yeast Gir2, which heterodimerizes with developmentally regulated GTP-binding protein 2 (DRG2), with which it is mutually stabilized in a manner similar to that of the yeast Gir2/Rbg2 complex [32, 33]. It is noteworthy that DRG2 is involved in G2/M regulation such that both the overexpression and knockdown of *DRG2* increase the number of cells in the G2/M phase [34, 35]. Therefore, we speculated that GCN1 might modulate the cell cycle through the DFRP2/DRG complex. In *DRG2*-knockdown HeLa cells, CDK1 and Cyclin B1 are decreased, while p21 is increased [34]. It is well known that the CDK1/Cyclin B1 complex induces G2/M transition and that p21 inhibits CDKs/cyclins [36, 37]. Therefore, we analyzed *Cdk1*, *Cyclin B1* and *p21* gene expression in the *Gcn1$^{ΔRWDBD}$* and *Gcn2* KO MEFs. Decreased *Cdk1* gene expression was observed in the *Gcn1$^{ΔRWDBD}$* MEFs, but not in the *Gcn2* KO MEFs, compared to the levels in the corresponding WT MEFs (Fig 6A and 6B). In contrast, an increase in *p21* gene expression was observed both in the *Gcn1$^{ΔRWDBD}$* and *Gcn2* KO MEFs. The increase in p21 and decrease in Cdk1 in the *Gcn1$^{ΔRWDBD}$* MEFs were also observed at the protein level, which was consistent with respective level of the mRNA (Fig 6C and quantified

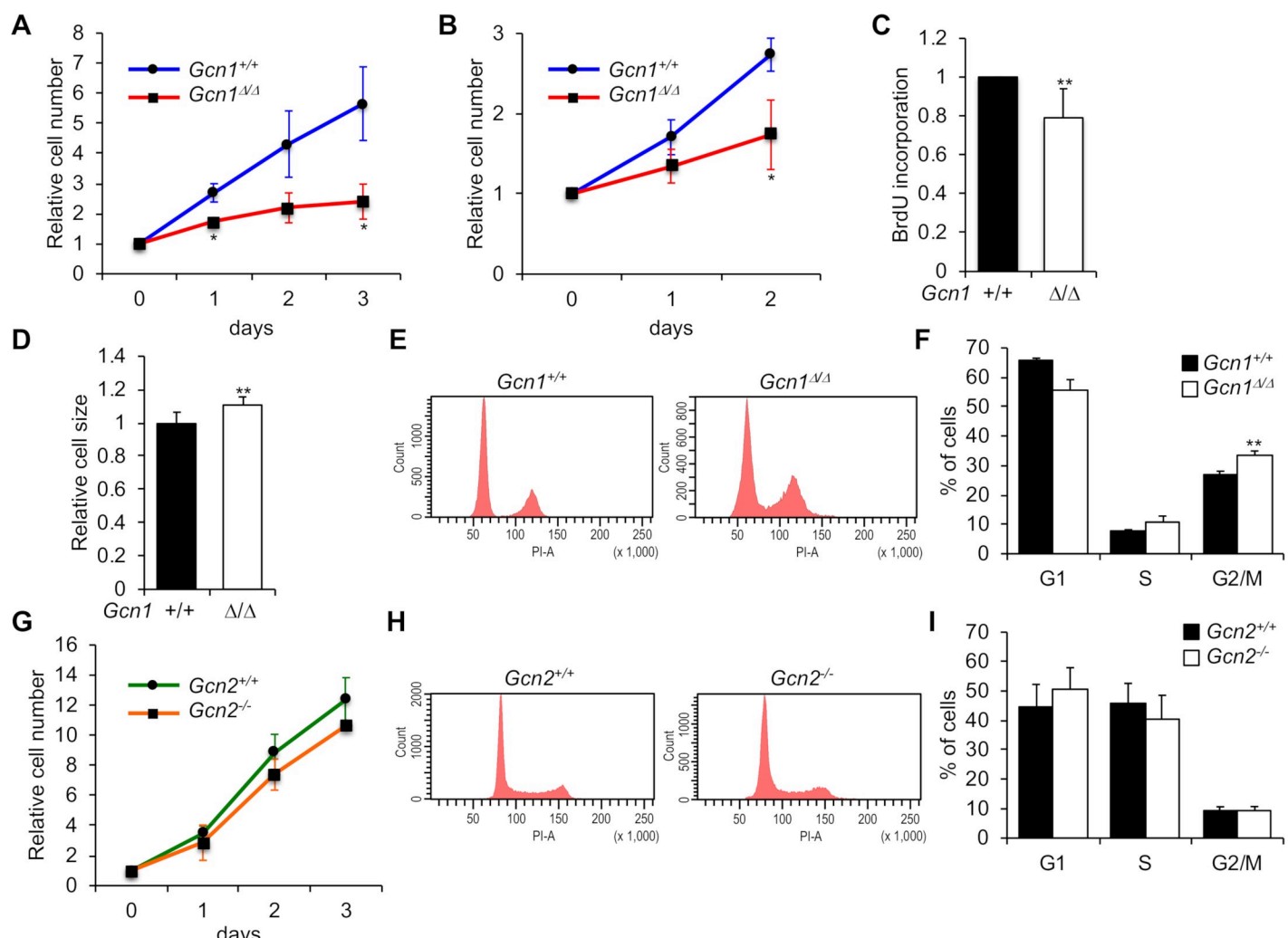

**Fig 5. *Gcn1^{ΔRWDBD}* MEFs exhibited reduced cell proliferation.** (A)(B)(G) After cells were cultured in IMDM for the indicated periods, the relative cell numbers of primary *Gcn1^{ΔRWDBD}* MEFs (A), immortalized *Gcn1^{ΔRWDBD}* MEFs (B) and immortalized *Gcn2* KO (*Gcn2^{-/-}*) MEFs (G) were counted and are shown with those of the corresponding WT (*Gcn1^{+/+}* or *Gcn2^{+/+}*) cells. The initial cell number was set to 1 and the results are shown as the relative folds±SD from multiple independent experiments ((A): N = 6, (B): N = 4, (C): N = 3). * $p < 0.05$, ** $p < 0.01$ compared with the WT (two tailed Student's *t*-test). (C) Cell proliferation in the primary WT (*Gcn1^{+/+}*) and *Gcn1^{ΔRWDBD}* MEFs was examined using BrdU incorporation. The results are presented as fold differences compared to the WT (*Gcn1^{+/+}*) MEFs from multiple independent experiments (N = 6). ** $p < 0.01$ compared with the WT (two tailed Student's *t*-test). (D) Relative cell sizes in the primary WT (*Gcn1^{+/+}*) and *Gcn1^{ΔRWDBD}* MEFs were measured using forward light scatter (FSC) of flow cytometry. The results are presented as fold differences compared to the WT (*Gcn1^{+/+}*) MEFs from multiple independent experiments (N = 7). ** $p < 0.01$ compared with the WT (two tailed Student's *t*-test). (E) The WT (*Gcn1^{+/+}*) and *Gcn1^{ΔRWDBD}* primary MEFs were stained by propidium iodide (PI) to assess the DNA contents in different phases of the cell cycle. The percentage of cells in each cell cycle was calculated (F). The results are presented as fold differences compared to those of the WT (*Gcn1^{+/+}*) MEFs from multiple independent experiments (N = 5). ** $p < 0.01$ compared with the WT (two tailed Student's *t*-test). (H) WT (*Gcn2^{+/+}*) and immortalized *Gcn2* KO (*Gcn2^{-/-}*) MEFs were stained by PI, and the percentage of cells in each cell cycle was calculated (I). Data are shown as means±SD from multiple independent experiments (N = 4).

in S10 Fig). Interestingly, Cyclin B1 protein levels were decreased in the *Gcn1^{ΔRWDBD}* MEFs compared to those in the WT MEFs (Fig 6C and quantified in S10 Fig), although the mRNA expression was comparable to that of the WT MEFs (Fig 6A). On the other hand, the Cyclin B1 protein level in the *Gcn2* KO MEFs was comparable to that in the control MEFs (Fig 6D and quantified in S10 Fig). Thus, it is likely that GCN1 modulates the cell cycle through the upregulation of Cyclin B1 and Cdk1. In addition, p21 inhibits the Cyclin B1/Cdk1 complex [38]. Therefore, increased p21 in the *Gcn1^{ΔRWDBD}* MEFs might be involved in Cyclin B1/Cdk1 inhibition.

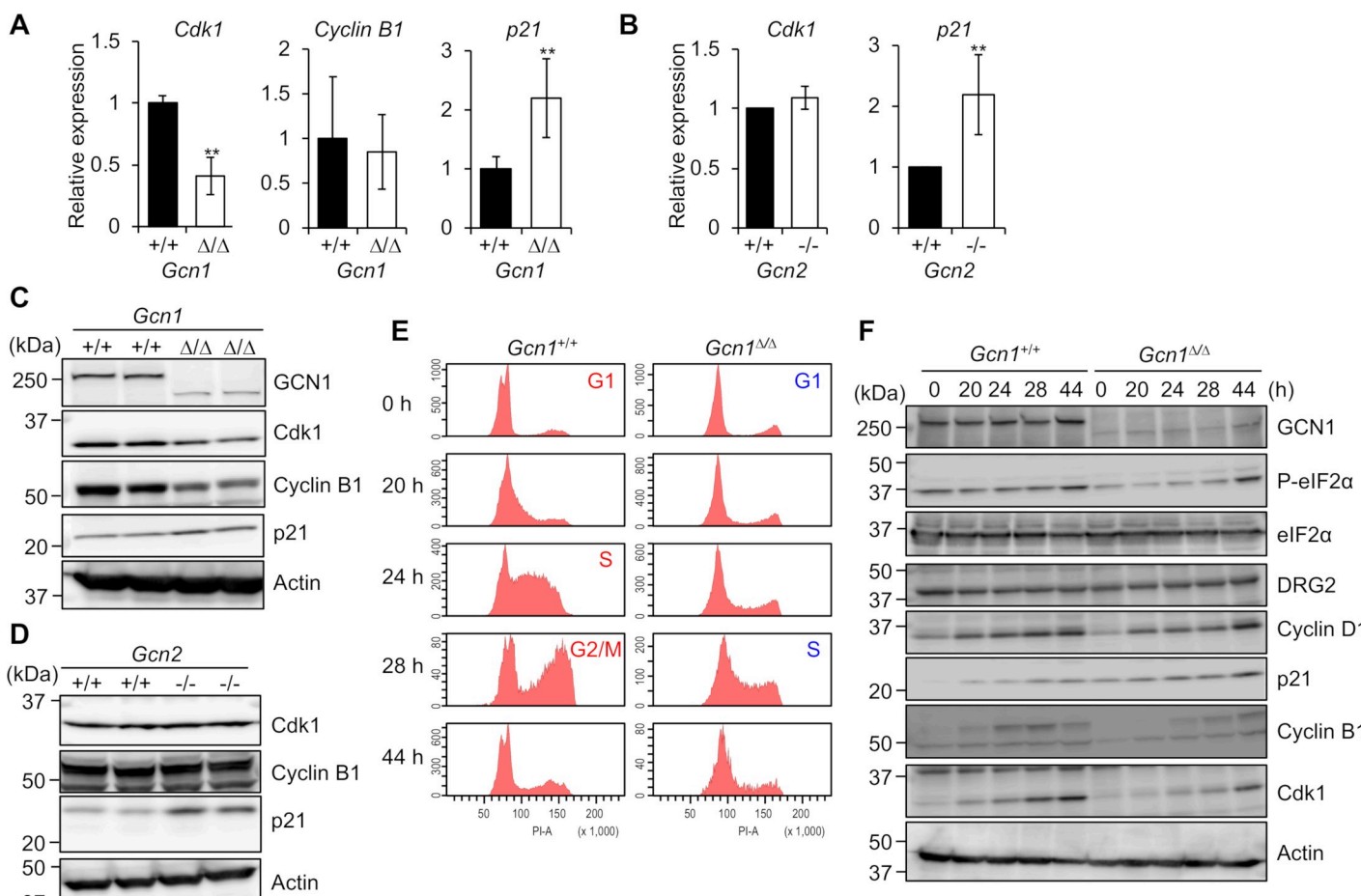

**Fig 6. Expression of cell cycle regulation factors in *Gcn1^ΔRWDBD^* MEFs.** (A) (B) Gene expression levels related to cell cycle progression in the *Gcn1^ΔRWDBD^* MEFs (A) and *Gcn2* KO (*Gcn2^-/-^*) MEFs (B) were quantified by RT-PCR. The value for wild-type cells was set to 1, and the results are shown as the relative folds±SD from multiple independent experiments ((A): N = 4, (B): N = 4). * $p < 0.05$, ** $p < 0.01$ compared with the WT (two tailed Student's *t*-test). (C) Whole cell proteins extracted from the WT (*Gcn1^+/+^*) and *Gcn1^ΔRWDBD^* MEFs were subjected to immunoblotting to detect GCN1, Cdk1, Cyclin B1, p21 and β-actin. (D) Whole cell proteins extracted from the WT (*Gcn2^+/+^*) and *Gcn2* KO (*Gcn2^-/-^*) MEFs were subjected to immunoblot analysis to detect Cdk1, Cyclin B1, p21 and β-actin. (E) Cell cycle analysis of the MEFs. WT (*Gcn1^+/+^*) and *Gcn1^ΔRWDBD^* MEFs were synchronized at G0-G1 by serum deprivation for 72 h. After serum stimulation at the indicated times, the cells were treated with PI, and the cell content was analyzed using flow cytometry. (F) Whole cell protein extracts from the WT (*Gcn1^+/+^*) and *Gcn1^ΔRWDBD^* MEFs obtained by the experiment in (E) were subjected to immunoblot analysis to detect GCN1, phosphorylated eIF2α (P-eIF2α), eIF2α, DRG2, Cyclin D1, p21, Cyclin B1, Cdk1 and β-actin.

Originally, p21 was reported to inhibit Cyclin E/Cdk2 and Cyclin D/Cdk4/6, which regulate the G1/S transition [39]. Therefore, we compared cell cycle progression between the *Gcn1^ΔRWDBD^* MEFs and the WT MEFs. The MEFs were synchronized to the G0/G1 phase by serum deprivation for 72 h and thereafter were released into the cell cycle by the replacement of the medium with fresh medium. Replenishment with fresh medium resulted in more WT MEFs in the S phase at 24 h, but at 28 h, fewer *Gcn1^ΔRWDBD^* MEFs were in the S phase compared to WT MEFs (Fig 6E and quantified in S11 Fig). These results suggest that the *Gcn1^ΔRWDBD^* MEFs are delayed at the G1/S transition. Accumulation of Cyclin B1 and Cdk1 was also observed in the *Gcn1^ΔRWDBD^* MEFs, although to a lesser extent than that in the WT MEFs (Fig 6F). Interestingly, p21 expression was higher in the *Gcn1^ΔRWDBD^* MEFs at every time point, suggesting that an increase in p21 might be one of the causes of the cell cycle delay. Notably, serum starvation-induced eIF2α phosphorylation was decreased in the *Gcn1^ΔRWDBD^* MEFs and was not recovered until 44 h after the medium was replaced (Fig 6F).

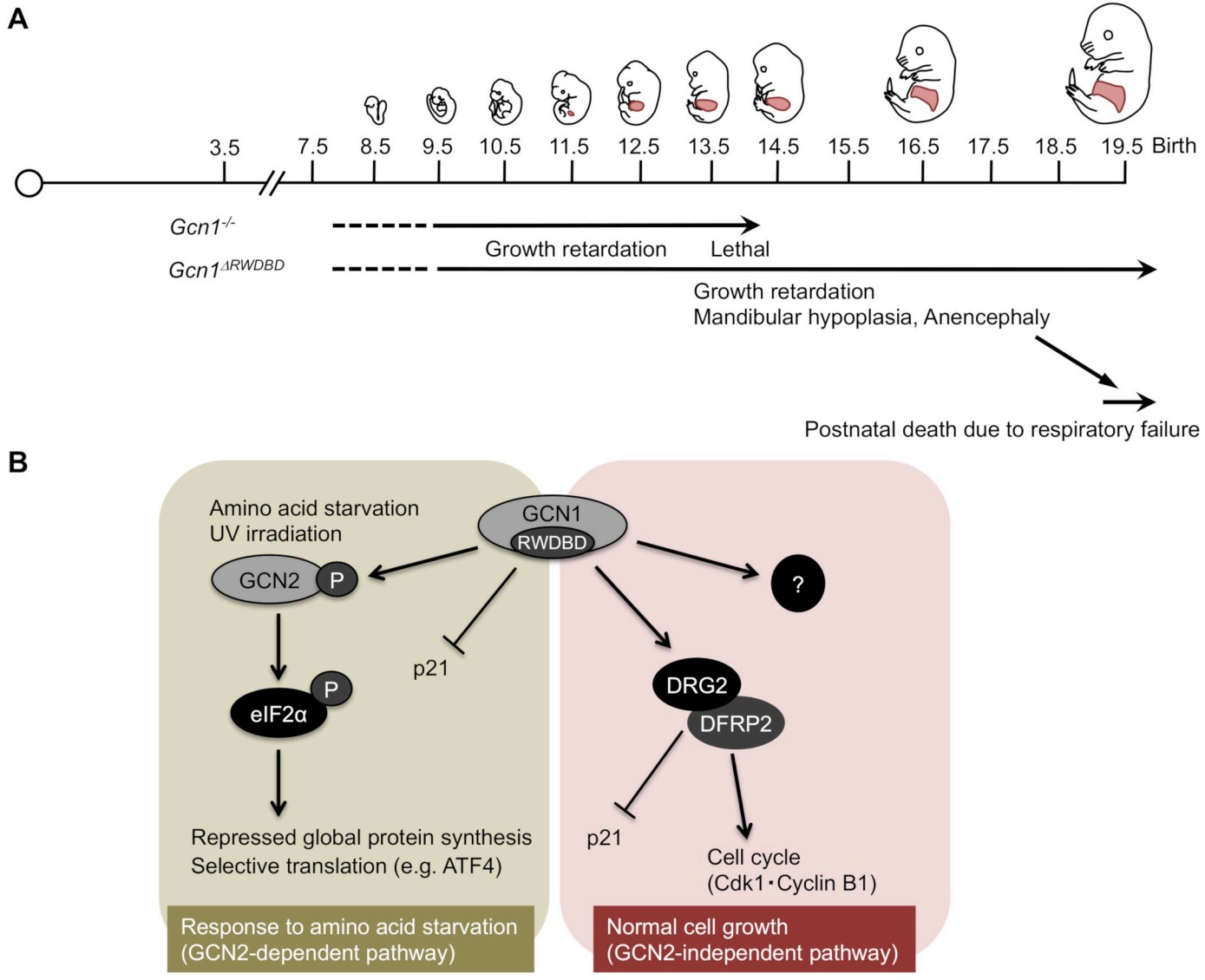

**Fig 7. Graphical summary.** (A) Summary of phenotypes of the *Gcn1* KO and *Gcn1^ΔRWDBD* embryos. (B) Schematic of the role of GCN1 in the control of cell proliferation and cell cycles.

## Discussion

In this study, we clarified, for the first time, that GCN1 is essential for embryonic development by generating two *Gcn1* mutant mouse lines (i.e., *Gcn1* KO and *Gcn1^ΔRWDBD* mice). We summarize the developmental defects observed in *Gcn1* KO and *Gcn1^ΔRWDBD* mice in Fig 7A. *Gcn1^ΔRWDBD* mice showed growth defects as early as E9.5 and died perinatally, likely due to respiratory failure caused by poor differentiation and insufficient lung inflation (Figs 1 and 2). Simple developmental delay, but not the absolute requirement for GCN1 RWDBD, during lung differentiation presumably underlies this phenotype, as prolonged gestation by progesterone administration at least partially rescued the lung differentiation defects and, consequently, the lethality observed immediately after the cesarean section. As *Gcn2* KO mice did not show defects in embryonic development and perinatal lethality [12], our results indicate that the role of GCN1 in embryonic development is GCN2-independent. This is fully consistent with our

*in vitro* results, which revealed that the *Gcn2* KO MEFs neither showed proliferative defects nor decreased Cdk1 and cyclin B1 protein levels, as observed in the *Gcn1*$^{\Delta RWDBD}$ MEFs (Fig 6). Interestingly, *GCN1* mutation in human was recently suggested to lead to fetal akinesia that is characterized by reduced or absent fetal movement, but often associate with multiple abnormalities including intra uterine growth retardation [40]. Although GCN1 protein levels were decreased to approximately 30–40% compared to those of the wild-type MEFs, it is not clear at this point to what extent the decrease in GCN1 protein levels contributes to the phenotypes of *Gcn1*$^{\Delta RWDBD}$ mice other than the lack of the RWDBD. *Gcn1* heterozygous KO MEFs expressed about half amount of mRNA and protein compared with WT MEFs (S12A–S12C Fig). And heterozygous *Gcn1* KO mice showed neither perinatal lethality nor embryonic growth defects (S12D Fig), suggesting that at least one-half the amount of GCN1 is sufficient for normal development.

Recent reports in *C. elegans* and *A. thaliana* have clarified the pleiotropic roles of GCN1, which are independent of GCN2. Of note, GCN1 acts with GCN20 independently of GCN2 to regulate translation that enhances the response to infection and mitochondrial dysfunction in Arabidopsis [25]. Consistent with the severe phenotype of *Gcn1* KO mice compared to *Gcn1*$^{\Delta RWDBD}$ mice, Arabidopsis *gcn1* mutants showed basic growth retardation that was more severe in the mutants that lacked an extensive portion of the sequence toward the GCN1 N-terminus. The importance of the central region of GCN1 spanning the GCN20 binding domain for apoptosis regulation was reported for *C. elegans* mutants [24]. Therefore, it is possible that the phenotypic difference between the *Gcn1* KO and *Gcn1*$^{\Delta RWDBD}$ mice is attributable to the GCN20 binding domain, since this domain is intact in the *Gcn1*$^{\Delta RWDBD}$ mice (Fig 1E). Considering that the *GCN1* mutation in *C. elegans* and *A. thaliana* caused growth defects, the role of GCN1 in developmental growth and cell cycle progression during the G2/M transition might be conserved among eukaryotes.

We observed growth defects and an increase in the G2/M population in both immortalized and primary *Gcn1*$^{\Delta RWDBD}$ MEFs (Fig 5A, 5B, 5E and 5F and S13 Fig). We also observed a delay in G1/S progression in the cell cycle-synchronized cells (Fig 6E and quantified in S11 Fig) and decreased incorporation of BrdU in the primary *Gcn1*$^{\Delta RWDBD}$ MEFs compared to the that in WT MEFs (Fig 5C). Although only a slight increase in apoptotic markers was observed in immortalized *Gcn1*$^{\Delta RWDBD}$ MEFs, we consider that the decrease in cell growth in the *Gcn1*$^{\Delta RWDBD}$ MEFs is mainly due to decreased cell proliferation but not to increased cell death. Although increase of *p21* mRNA and p21 protein levels were observed in *Gcn2* KO MEFs, we did not observe proliferation defects of the cells. One caveat in this study is that we used *Gcn2* KO MEF that was immortalized by large T antigen [41]. Therefore, proliferation defects can be rescued by the expression of large T antigen. However, we speculate that *Gcn2* KO cells do not show proliferation defects *in vivo* as *Gcn2* KO mice did not show overt growth defects during embryonic development.

It is known that yeast Gir2 binds to Gcn1 via the Gir2 RWD domain. Gir2 heterodimerizes with Rbg2 and is necessary for maximum proliferation under AAS [32]. Under AAS, the binding of Gcn1 with Gir2/Rbg2 is increased, and the Gir2/Rbg2 complex is stabilized in a GTP-dependent manner. DFRP2, the human homolog of Gir2, heterodimerizes human DRG2 and they stabilize each other in a manner similar to Gir2 and Rbg2 in yeast [33] (Fig 7B). The high conservation of both amino acid sequences and the mutual stabilization mechanism between the yeast Gir2/Rbg2 and human DFRP2/DRG2 complexes suggest their functional importance in cell growth. Jang *et al.* showed that *DRG2*-deficient HeLa cells showed a defect in cell proliferation and G2/M arrest associated with p21 induction and a decrease in Cdk1 and Cyclin B1 expression that were observed in the *Gcn1*$^{\Delta RWDBD}$ MEFs (Fig 6A and 6C). The G2/M arrest observed in the *Gcn1*$^{\Delta RWDBD}$ MEFs might reflect a disrupted DRG2/DFRP2 function [34] (Fig

7B). The only difference in this respect is that both the *Cyclin B1* mRNA expression and the protein levels were decreased in *DRG2*-knockdown HeLa cells, while only the cyclin B1 protein levels were decreased in the *Gcn1$^{\Delta RWDBD}$* MEFs (Fig 6A and 6C and quantified in S10 Fig). It is well known that *Cyclin B1* mRNA is upregulated in the G2/M phase and that the Cyclin B1 protein is degraded by the APC/C-mediated ubiquitin-proteasome pathway before the onset of mitosis [42]. Therefore, the Cyclin B1 protein level might be decreased by the premature activation of APC/C. It is of note that *Drg2* deficient mice were just recently reported and that they show embryonic growth retardation, but survived to the adulthood [43]. The observation strongly supports our contention that defect of DRG2 might be the cause of growth retardation in *Gcn1$^{\Delta RWDBD}$* mice.

Notably, *Ola1* KO mice show strikingly similar phenotypes as those of *Gcn1$^{\Delta RWDBD}$* mice (i.e., embryonic growth retardation and perinatal lethality due to respiratory failure) [44]. OLA1 belongs to an Obg-family GTPase and reduces ternary complex formation by binding to eIF2 and hydrolyzing GTP, which leads to the activation of ISR [45]. Ding *et al.* showed that *Ola1* KO mice exhibit growth retardation and high levels of embryonic lethality due to the decreased cell proliferation associated with increased *p21* mRNA translation. They also showed that the increased *p21* mRNA translation in the *Ola1* KO MEFs is reversed by an eIF2 phosphatase inhibitor, salubrinal, indicating that translation of *p21* mRNA is sensitive to the amount of ternary complex [44]. Although both OLA1 and GCN1 can mediate stress-induced decreases in ternary complex formation, the reasons for similarity should be clarified in the future. In contrast to their similar *in vivo* phenotypes, *Ola1* KO MEFs show G1/S arrest associated with the decrease in Cyclin D1 and E1, but not Cyclin B1, which differs from the phenotypes of the *Gcn1$^{\Delta RWDBD}$* MEFs. However, it is noteworthy that the *Ola1* KO MEF population is also increased in G2/M. The detailed molecular connections between OLA1 and GCN1 need to be explored in future studies.

The levels of the p21 protein and the *p21* mRNA were increased in the *Gcn2* KO MEFs and in the *Gcn1$^{\Delta RWDBD}$* MEFs (Fig 6A–6D and quantified in S10 Fig). Both increased transcription and translation could potentially contribute to the increase in p21 protein levels in these cells. As described above, p21 translation is increased by an increase in the ternary complex. Indeed, the basal level of eIF2α phosphorylation appeared to decrease in both *Gcn1$^{\Delta RWDBD}$* and *Gcn2* KO MEFs (Fig 4). On the other hand, Nakamura *et al.* recently reported that GCN2 localizes at the nucleolus and that *p21* mRNA transcription is increased by *GCN2* knockdown in a p53-dependent manner [46]. p53 activity is regulated by the signals from the nucleolus [47] and the transcripts in the nucleolus such as ribosomal 5S RNA are proposed to activate p53 [48]. Nakamura *et al.* showed that accumulated small RNAs including 5S RNA lead to p53 activation in *GCN2* knockdown cancer cells indicating that GCN2 may inhibit RNA polymerase III-dependent transcription including 5S RNA. Although we failed to detect p53 accumulation in *Gcn1$^{\Delta RWDBD}$* MEFs, but it is an interesting possibility that GCN1 regulates GCN2 localization at the nucleolus and its function. As *p21* mRNA is also increased by the *DRG2* knockdown in HeLa cells [42], we surmise that both GCN2 and DRG2 are involved in *p21* mRNA upregulation in *Gcn1$^{\Delta RWDBD}$* MEFs.

In conclusion, we confirmed that GCN1 is required for the GCN2-dependent response to stress such as that induced by AAS and UV radiation, which is consistent with a previous report [22] (Fig 7B). Furthermore, we clarified that GCN1 is necessary for cell proliferation and development and that the GCN1 action is GCN2-independent. Thus, we propose that GCN1 is a hub protein of cellular signaling that integrates cellular stress responses and growth. This study established a molecular basis for the future clarification of mammalian GCN1 function (Fig 7B).

## Materials and methods

### Ethics statement

Mice were maintained in temperature- and humidity-controlled rooms on a 12-h light-dark cycle. All mouse experiments were approved by the Committee for the Ethics of Animal Experimentation of Hirosaki University (#M14040) and carried out according to the Guideline for Animal Experimentation of Hirosaki University.

### Generation of *mutant* mouse by CRISPR/Cas9 system

**Gcn1 KO mice.** *Gcn1* KO mice were generated similarly to the *Gcn1$^{ΔRWDBD}$* mice. To construct a Cas9/sgRNA expression vector, the following oligonucleotide DNAs were used: for exon 2 KO (targeted to introns 1 and 2), 5'-TGC TGC TGA TGT GAG TGC GG-3' and 5'-GCC TGT AGG GAC TGT TCT CG-3'). Genotypes were determined by PCR with primer sets for the WT allele: 5'-AGA TGG TAG CAG GTG GCG TCC AG-3' and 5'-GGG ATG GAA GGT AGG CCG GCT G-3', and for the deleted allele: 5'-TGC TCT TGT GGG CCT GTG CCA-3' and 5'-GGG ATG GAA GGT AGG CCG GCT G-3'.

**Gcn1$^{ΔRWDBD}$ mice.** To construct a Cas9/single-guide RNA (sgRNA) expression vector, oligonucleotide DNAs (for exons 46–53 KO (targeted to introns 45 and 53): 5'-AGA GCC CTC ACC TAT CCT AT-3' and 5'-TCA AAT CCT GGC ACC GA-3') were annealed and inserted into the pX330 vector (Addgene plasmid # 42230). The pX330 vector was injected into the pronuclei of C57BL/6 fertilized eggs. F0 mice were genotyped for the deletion between exons 46 and 53 of the *Gcn1* gene due to a double-strand break in both introns 45 and 53 and nonhomologous end-joining (NHEJ). F0 mice were checked for the absence of the Cas9 transgene and off-target effects. F0 mice were mated with the C57BL/6 mice to obtain F1 offspring. Genotypes were determined by PCR with the following primer sets for the WT allele: 5'-TGT CCC TTA GTG TGT TAG ATT CC-3' and 5'- TGC AGC TGC TCA AAG GTC TTG G-3', and the following primer set for the deleted allele: 5'-TGT CCC TTA GTG TGT TAG ATT CC-3' and 5'-AGT CAA GCT GAC TCT TGA CTG C-3'. To avoid using mice with unexpected off-target effects, F1 mice were backcrossed at least 8 times with WT C57BL/6 mice purchased from CLEA Japan Inc. and used for analyses.

### Administration of progesterone

*Gcn1$^{+/ΔRWDBD}$* pregnant mice at E18.5 were administered 4 mg of progesterone by subcutaneous (*s.c.*) injection. Progesterone-injected mice could maintain pregnancy, and embryos at E20.5 were delivered by cesarean section.

### Quantitative RT-PCR (q-RT-PCR)

For q-RT-PCR, total RNA was prepared by using TRIzol RNA Isolation Reagents (ThermoFisher, Tokyo, Japan) following the manufacturer's protocol. cDNA was synthesized from total RNA using PrimeScript Reverse Transcriptase (TaKaRa Bio Inc., Shiga, Japan). Real-time PCR was performed using primer sets in Table 4, TB Green Premix Ex Taq II (Takara Bio Inc) and a CFX96 thermal cycler (Bio-Rad, Richmond, CA).

### Cell culture

Human cervical carcinoma HeLa cells were maintained in high glucose Dulbecco's modified Eagle medium (DMEM, Sigma-Aldrich, St. Louis, MO) containing 10% fetal bovine serum, penicillin-streptomycin (PS) (100 U/ml-0.1 mg/ml, ThermoFisher) and 2 mM L-glutamine (Sigma-Aldrich). MEFs were prepared from individual embryos at E13.5-E15.5. After the head

**Table 4. Primer sets used to detect indicated genes by q-RT-PCR.**

| Gene | Forward (5'-3') | Reverse (5'-3') |
|---|---|---|
| CycA | AAGACTGAATGGCTGGATGG | AGCTGTCCACAGTCGGAAAT |
| Aqp5 | TCTTGTGGGGATCTACTTCACC | TGAGAGGGGCTGAACCGAT |
| Sp-c | ATGGACATGAGTAGCAAAGAGGT | CACGATGAGAAGGCGTTTGAG |
| Asns | GGCCCTTGTTTAAAGCCATGA | AAGGGAGTGGTGGAGTGTTTT |
| Psat1 | CAGTGGAGCGCCAGAATAGAA | CCTGTGCCCCTTCAAGGAG |
| Cdk1 | AGGTACTTACGGTGTGGTGTAT | CTCGCTTTCAAGTCTGATCTTCT |
| Cyclin B1 | CTTGCAGTGAGTGACGTAGAC | CCAGTTGTCGGAGATAAGCATAG |
| p21 | CCTGGTGATGTCCGACCTG | CCATGAGCGCATCGCAATC |
| Gcn1 (S1A–S1C Fig) | CTCTGCTGGAGACACTCAGC | GTGAAGGTGGTCTGAAGCTG |
| Gcn1 (S12A Fig) | AACAGCCAGTGTGAAGGAGC | CGATGCAGTGTCAAGCAGAAC |
| Gcn2 | TACTTTGCGATGAACTCCAGAGA | GCTCAGGTGTGTAGCCAGAG |
| Impact | ACGCGCAGACTTATCGAACA | TCTGGGTCTGGCTCGGTTAT |

and internal organs were removed, the torso was minced and dispersed in 0.05% trypsin-EDTA. MEFs were maintained in Iscove's modified Dulbecco's medium (IMDM, Sigma-Aldrich) containing 10% fetal bovine serum (FBS, Life Technologies, Carlsbad, CA), PS and 2 mM L-glutamine (Sigma-Aldrich). The generation of $Gcn2^{-/-}$ MEFs and WT cells was previously described [12]. For AAS experiments, MEFs were maintained in DMEM lacking specific amino acids (custom-made by Cell Science & Technology Institute, Sendai, Japan) containing 10% dialyzed FBS, 4.5 g/L glucose and PS. MEFs were treated either with $H_2O_2$ (Kanto Chemical, Tokyo, Japan), Tm (Wako Pure Chemicals, Osaka, Japan) or MG132 (Peptide Institute, Osaka, Japan). MEFs were transfected by Poly(I:C) by using a Lipofectamine 2000 (Thermo-Fisher). The cells were maintained at 37˚C in a 5% $CO_2$ incubator.

## Immunofluorescence

HeLa cells were fixed with 4% paraformaldehyde (PFA)/PBS and permeabilized with 0.25% Triton X-100. The following antibodies were used: GCN1 (Abcam, Cambridge, UK), calnexin (BD Biosciences, Tokyo, Japan), pyruvate dehydrogenase (PDH) subunit E1 antibody (Invitrogen, Carlsbad, CA). Detection was performed using Alexa Fluor 488 for GCN1 and Alexa Fluor 594 for calnexin and PDH. 4,6-Diamidino-2-phenylindole (DAPI) was used for nuclear counterstaining. The fluorescent images were observed using the C1si confocal imaging system (Nikon, Tokyo, Japan).

## Immunoblot analysis

The cytosolic or nuclear extracts were prepared as previously described [49].

The protein samples were boiled in Laemmli buffer and electrophoresed under reducing conditions by SDS-PAGE. Protein concentrations were determined using a bicinchoninic acid (BCA) protein assay kit (Pierce Biotechnology, Rockford, IL).

Proteins were transferred to polyvinylidene difluoride (PVDF) membranes (Millipore) and subjected to immunoblot analysis using primary antibodies against GCN1 (Abcam), GCN2 (Cell Signaling Technology, Danvers, Massachusetts), phospho-Thr899 GCN2 (Abcam), eIF2α (Cell Signaling Technology), phospho-Ser51 eIF2α (Cell Signaling Technology), DRG2 (Abcam), Cdk1 (Cell Signaling Technology), cyclin B1 (Abcam), cyclin D1 (Santa Cruz, Dallas, Texas), p21 (Santa Cruz), PARP (Cell Signaling Technology), Caspase-3 (Cell Signaling Technology) and β-actin (Sigma) and HRP-labeled secondary antibodies. Immunoreactive bands

were visualized by using ImmunoStar chemiluminescent reagent (Wako Pure Chemicals, Tokyo, Japan) and intensities were measured by using Image J software 1.52q (NIH, Bethesda, MD).

## UV irradiation

MEFs were washed twice with PBS, and the PBS was removed from the plates and cells. The cells were irradiated at a dose of 80 J/m$^2$ and 254 nm by using a UV cross-linker (CL-1000, UVP Inc.). After UV irradiation, fresh medium was added to each plate.

## Histological analysis

Lungs were fixed with 10% formalin (Mildform 10 N; Wako Pure Chemicals) and embedded in paraffin using standard procedures. Sections (5 μm) were stained with hematoxylin-eosin.

## Cell cycle analysis

The cells were collected, washed with PBS, and fixed in 70% ethanol at -20˚C for at least 2 h. The fixed cells were washed with PBS, stained with a Tali cell cycle kit (containing propidium iodide (PI) and RNase A) (ThermoFisher) for 30 min, and analyzed using a Canto II flow cytometer (BD Biosciences). The cell sizes were examined using a forward light scatter (FSC) parameter. Data were analyzed using the FACSDiva Software (BD Biosciences), ModFit LT3.2 Software (BD Biosciences) and FlowJo software (BD Biosciences).

## Cell viability analysis

MEFs were plated on 96-well plates at a density of $5.0 \times 10^4$ cells/well. The number of viable cell numbers was quantified with a Cell Counting Kit-8 (Dojindo Molecular Technologies, Kumamoto, Japan) according to the manufacturer's protocol.

## BrdU incorporation

MEFs were plated on 96-well plates and incubated for 24 h. The cells were labeled with BrdU for 2 h, and the incorporation of BrdU was analyzed using a BrdU cell proliferation kit (Merck Millipore, Darmstadt, Germany) according to the manufacturer's protocol. Each value was normalized by cell number using a Cell Counting Kit-8.

## Metabolic labeling

Metabolic labeling was examined as previously reported [49].

## Senescence-associated β-galactosidase staining

MEFs were plated on glass bottom dish and incubated for 24 h in the culture medium and β-galactosidase staining was performed with a Cellular Senescence Assay Kit (Cell Biolabs Inc., San Diego, CA) according to the manufacturer's protocol.

## Statistical analysis

Two tailed Student's *t*-tests or one-way ANOVA with Bonferroni post hoc test were used for analyses of differences between groups. The data are presented as means with standard deviation (SD) values. Statistical significance was considered as P<0.05.

## Supporting information

**S1 Fig. Expression patterns of *Gcn1* mRNA in embryos.** (A) *Gcn1* expression in WT embryos at each stage was quantified by RT-PCR. The values for E9.5 were set to 1, and the results are shown as relative means±SD from multiple independent animals (N = 4). (B) *Gcn1* expression in each part of the WT embryos at E14.5 was quantified by RT-PCR. The value for the head was set to 1, and the results are shown as relative means±SD from multiple independent animals (E9.5-E14.5: N = 4). (C) *Gcn1*, *Gcn2*, *Impact* expression of each organ of the WT embryos at E18.5 was quantified by RT-PCR. The value for the brain was set to 1, and the results are shown as relative means±SD from multiple independent animals (N = 3).
(TIF)

**S2 Fig. Developmental abnormalities that observed in *Gcn1^{ΔRWDBD}* embryos.** (A) Representative pictures of the embryos at E14.5. Scale bar: 5 mm. No.155 of *Gcn1^{ΔRWDBD}* embryo showed abnormalities in the head or had an anencephaly-like phenotype. (B) Representative pictures of the embryos at E20.5. No.260 of *Gcn1^{ΔRWDBD}* embryo was dead at the time of cesarean section and showed abnormalities in the head or had an anencephaly-like phenotype. (C) Pictures of face of *Gcn1^{ΔRWDBD}* embryo at E20.5 in S2B Fig. No.259 and 260 of *Gcn1^{ΔRWDBD}* embryos showed mandibular hypoplasia and exophthalmos/hypoplasia of the eyelid.
(TIFF)

**S3 Fig. Localization of GCN1 to the cytosol.** (A) Immunofluorescence analysis of GCN1 in HeLa cells. GCN1 localization is shown in green, and nuclear DAPI staining is shown in blue. The merged images are also shown. (B)(C) Double immunofluorescence staining of GCN1 (green) and calnexin (red) (B) or PDH (red) (C) in HeLa cells. Nuclear DAPI staining is shown (blue). The merged images are also shown. (D) HeLa cells were fractionated into cytosol (C), nuclear (N) and whole cell (W) fractions and subjected to immunoblot analysis to detect GCN1, Lamin B and β-actin. (E) MEFs were fractionated into cytosol (C), nuclear (N) and whole cell (W) fractions and subjected to immunoblot analysis to detect GCN1, α-Tubulin and Lamin B. Equal amounts of proteins were subjected to SDS-PAGE.
(TIFF)

**S4 Fig. Metabolic labeling of newly synthesized proteins.** (A) De novo synthesized proteins in the *Gcn1^{+/+}* and *Gcn1^{ΔRWDBD}* MEFs were measured using L-azidohomoalanine (AHA). (B) Protein levels were also confirmed by protein staining on the same membrane.
(TIFF)

**S5 Fig. GCN1 is necessary for GCN2-mediated ATF4 activation.** (A) The data in Fig 3B was quantified and shown. The value for the WT control was set to 1, and the results are shown as relative means±SD from multiple independent experiments (N = 3). (B) The replicate of Fig 3D was shown. The WT (*Gcn1^{+/+}*) and *Gcn1^{ΔRWDBD}* MEFs were exposed to leucine (Leu), methionine (Met), serine (Ser) or cystine (Cys) starvation for 4 h or cultured in the control (Ctrl) medium and cells were fractionated into cytosol, nuclear fractions and subjected to immunoblot analysis to detect the phosphorylated GCN2 (P-GCN2), GCN2, phosphorylated eIF2α (P-eIF2α), eIF2α, HSP90, ATF4 and Lamin B.
(TIFF)

**S6 Fig. GCN1 and GCN2 dependency in response to UV exposure.** (A) The data in Fig 4A was quantified and shown. The value for the WT control cells was set to 1, and the results are shown as relative means±SD from multiple independent experiments (N = 3). (B) The data in Fig 4B was quantified and shown. The value for the WT control cells was set to 1, and the

results are shown as relative means±SD from multiple independent experiments (N = 3).
(TIFF)

**S7 Fig. The role of GCN1 in eIF2α phosphorylation by HRI, PERK and PKR.** (A)(B) The WT and $Gcn1^{\Delta RWDBD}$ (A) or $Gcn2$ KO ($Gcn2^{-/-}$) MEFs (B) were treated by $H_2O_2$ for 1 hour and subjected to immunoblot to detect phosphorylated GCN2 (P-GCN2), GCN2, phosphorylated eIF2α (P-eIF2α), eIF2α and β-actin. (C)(D) The WT and $Gcn1^{\Delta RWDBD}$ (C) or $Gcn2$ KO ($Gcn2^{-/-}$) MEFs (D) were treated by 10 μM MG132 (MG) or 2 μg/mL Tm for 1 hour and subjected to immunoblot to detect phosphorylated GCN2 (P-GCN2), GCN2, phosphorylated eIF2α (P-eIF2α), eIF2α and β-actin. (E) The WT and $Gcn1^{\Delta RWDBD}$ MEFs were treated by 2 μg/mL Tm for 16 hours, and the mRNA levels of the ATF4 target genes *Asns* and *Psat1* were quantified by RT-PCR. The value for WT control cells was set to 1, and the results were shown as the relative folds±SD from multiple independent experiments (N = 4). * $p<0.05$, ** $p<0.01$ compared with the WT (two tailed Student's *t*-test). (F)(G) The WT and $Gcn1^{\Delta RWDBD}$ (F) or $Gcn2$ KO ($Gcn2^{-/-}$) MEFs (G) were transfected with Poly(I:C) and incubated for 4 hour and subjected to immunoblot to detect phosphorylated GCN2 (P-GCN2), GCN2, phosphorylated eIF2α (P-eIF2α), eIF2α.
(TIF)

**S8 Fig. Detection of apoptosis in $Gcn1^{\Delta RWDBD}$ MEFs.** (A) Whole cell proteins extracted from WT ($Gcn1^{+/+}$) and primary (A) or immortalized (B) $Gcn1^{\Delta RWDBD}$ MEFs were subjected to immunoblot analysis to detect PARP, Caspase-3 and β-actin. Intact and cleaved forms of PARP and Caspase-3 are indicated with filled and open arrowheads, respectively. WT MEFs were treated with 2 μM doxorubicin (DXR) for 16 h and loaded as a positive control during the analysis of apoptotic cells. (C) The data in S8B Fig was quantified and shown. The results are shown as relative means±SD from multiple independent experiments (N = 4).
(TIFF)

**S9 Fig. Analysis of senescence marker, β-galactosidase in $Gcn1^{\Delta RWDBD}$ MEFs.** Primary WT ($Gcn1^{+/+}$) and $Gcn1^{\Delta RWDBD}$ MEFs were subjected to β-galactosidase staining and representative data was shown. Independent mouse lines were used and analyzed. The arrowheads indicate β-galactosidase positive cells.
(TIFF)

**S10 Fig. The expression of Cdk1, Cyclin B1 and p21 in $Gcn1^{\Delta RWDBD}$ and $Gcn2$ KO MEFs.** The data in Fig 6C and 6D was quantified and shown. The value for the WT was set to 1, and the results are shown as relative means±SD from multiple independent experiments (N = 3). ** $p<0.01$ compared with the WT (two tailed Student's *t*-test).
(TIF)

**S11 Fig. Cell synchronization by serum deprivation using $Gcn1^{\Delta RWDBD}$ MEFs.** The data in Fig 6E was quantified and shown. The results are shown as relative means±SD from multiple independent experiments (28 h: N = 2, 0 h, 20 h, 24 h and 44 h: N = 3). * $p<0.05$ compared with the WT (two tailed Student's *t*-test) (statistical analysis was not performed at 28 h).
(TIFF)

**S12 Fig. GCN1 expression analysis and cell growth in heterozygous $Gcn1$ KO MEFs.** (A) $Gcn1$ expression levels in the WT ($Gcn1^{+/+}$) and heterozygous $Gcn1$ KO MEFs were quantified by RT-PCR. Primers (spanning exon 2 to 3) were designed to detect WT mRNA but not mRNA from $Gcn1$ KO allele. The value for wild-type cells was set to 1, and the results are shown as the relative folds±SD from multiple independent experiments (N = 3). ** $p<0.01$ compared with the WT (two tailed Student's *t*-test). (B)(C) Whole cell protein extracted from

the WT ($Gcn1^{+/+}$) and heterozygous $Gcn1$ KO MEFs was subjected to immunoblot analysis to detect GCN1 and β-actin. The data in S12B Fig was quantified and shown in (C). The value for wild-type cells was set to 1, and the results are shown as the relative folds±SD from multiple independent experiments (N = 3). ** $p < 0.01$ compared with the WT (two tailed Student's $t$-test). (D) After cells were cultured in IMDM for the indicated periods, the relative cell numbers of primary heterozygous $Gcn1$ KO MEFs were counted and are shown with the corresponding WT ($Gcn1^{+/+}$) cells. The initial cell number was set to 1 and the results are shown as the relative folds±SD from multiple independent experiments (N = 3). * $p < 0.05$ compared with the WT (two tailed Student's $t$-test).
(TIF)

**S13 Fig. Cell cycle analysis in $Gcn1^{ΔRWDBD}$ primary MEFs.** The WT ($Gcn1^{+/+}$) and $Gcn1^{ΔRWDBD}$ immortalized MEFs were stained by propidium iodide (PI) to assess the DNA contents in different phases of the cell cycle and the percentage of cells in each cell cycle was calculated. The results are presented as fold differences compared to those of the WT ($Gcn1^{+/+}$) MEFs from multiple independent experiments (N = 6). ** $p < 0.01$ compared with the WT (two tailed Student's $t$-test).
(TIFF)

## Acknowledgments

We thank Drs. Masanobu Morita and Fumiki Katsuoka for critical advice. We also thank Siori Osanai, Fumiko Tsukidate, Yuko Tsushima, Michiko Nakata, Ayano Ono, Shun Igarashi and Dr. Ryo Ito for technical assistance and laboratory members for useful discussions.

## Author Contributions

**Conceptualization:** Hiromi Yamazaki, Shuya Kasai, Junsei Mimura, Peng Ye, Atsushi Inose-Maruyama, Tsubasa Sato, Ken Itoh.

**Data curation:** Hiromi Yamazaki.

**Formal analysis:** Hiromi Yamazaki.

**Funding acquisition:** Hiromi Yamazaki, Ken Itoh.

**Investigation:** Hiromi Yamazaki, Shuya Kasai, Junsei Mimura, Peng Ye, Atsushi Inose-Maruyama, Kunikazu Tanji, Koichi Wakabayashi, Seiya Mizuno, Fumihiro Sugiyama, Satoru Takahashi, Tsubasa Sato, Taku Ozaki, Douglas R. Cavener, Masayuki Yamamoto, Ken Itoh.

**Resources:** Seiya Mizuno, Fumihiro Sugiyama, Satoru Takahashi, Douglas R. Cavener.

**Supervision:** Ken Itoh.

**Writing – original draft:** Hiromi Yamazaki, Ken Itoh.

**Writing – review & editing:** Shuya Kasai, Junsei Mimura, Peng Ye, Atsushi Inose-Maruyama, Kunikazu Tanji, Koichi Wakabayashi, Seiya Mizuno, Fumihiro Sugiyama, Satoru Takahashi, Tsubasa Sato, Taku Ozaki, Douglas R. Cavener, Masayuki Yamamoto.

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
