## [Decision Letter · Decision Letter 0]

23 Sep 2019

Dear Dr Itoh,

Thank you very much for submitting your Research Article entitled 'Ribosome binding protein GCN1L1 regulates the cell cycle and cell proliferation and is essential for the embryonic development of mice' to PLOS Genetics. Your manuscript was fully evaluated at the editorial level and by three expert colleagues. All three found the key claim of the paper - that mammalian GCN1 has important functions distinct from its role in contributing to GCN2 activation (defined by the conspicuous and non-overlapping phenotypes associated with GCN1L1 and GCN2 l.o.f. mutation in mice) - to be well supported experimentally and of sufficient interest to merit publication in PLOS Genetics.

The reviewers also had some substantive criticisms, noted in their reports below.

One important point relates to the benefit that would arise from a more convincing demonstration that lack of GCN1 interferes with the activation of ISR markers in AA deprived cell (reviewers 1 & 3 allude to the measurement of ATF4 status in stressed wild type and GCN1 mutant cells, but any robust ISR marker could serve the purpose of this experiment). Related to that, reviewer 1 also wonders if it might be possible to explore the specificity of GCN1 for the activation of GCN2 (in contrast to any other eIF2a kinase). It may be possible to compare the activation of a robust ISR target (e.g. CHOP) in AA deprived or tunicamycin-treated WT and GCN1 mutant cells. This would report on the selectivity (if any) of GCN1 for GCN2 (the mediator of the ISR in AA starved cells) over PERK (the mediator of the ISR in tunicamycin-treated cells).

Based on the reviews, we hope to receive a revised version of the paper that addresses the key concerns of the reviewers. 

Your revisions should address the specific points made by each reviewer. We will also require a detailed list of your responses to the review comments and a description of the changes you have made in the manuscript.

Please aim to resubmit within the next 60 days, unless it will take extra time to address the concerns of the reviewers, in which case we would appreciate an expected resubmission date by email to plosgenetics@plos.org.

[LINK]

We are sorry that we cannot be more positive about your manuscript at this stage. Please do not hesitate to contact us if you have any concerns or questions.

Yours sincerely,

David Ron

Guest Editor

PLOS Genetics

Gregory Barsh

Editor-in-Chief

PLOS Genetics

**Comments to the Authors:**

Reviewer #1: In this article authors have produced, by using CRISPR/Cas9 technique, the first mice devoid of the function of GCN1L1 protein, the mammalian homolog of budding yeast protein Gcn1. Gcn1 has been shown to be non-essential for yeast cells, but necessary for the activation of the stress responsive eIF2α kinase Gcn2, also present in mammalian cells. Authors have obtained two types of mutant mice, one with no expression of GCN1L1, and the other expressing a truncated form of the protein which is supposed to be unable to interact with its main target GCN2. Interestingly, and unlike the GCN2 KO, both mice present a severe phenotype of growth retardation at the embryonic stage mainly affecting pulmonary development and maturation, suggesting an essential and GCN2-independent role for GCN1L1 in mammals.

Consequently, cells derived from these mice show diminished growth and GCN2-dependent stress response, but also additional cell cycle abnormalities absent in GCN2 KO cells.

The manuscript is well organized and clearly written, and, in general, data and analyses support the conclusions. Nevertheless, below there are a series of concerns:

Introduction

Ref. #5, studying PKR sensitivity to viral inhibitors, does not illustrate the authors’ statement “GCN2 might be the most ancient elF2α kinase in eukaryotes found in yeasts, plants and mammals, and budding yeast Saccharomyces cerevisiae (S. cerevisiae) possess Gcn2 as the sole elF2α kinase (5)”

Ref. #9 is a paper done by using the fission yeast Schizosaccharomyces pombe (S. pombe) instead of S. cerevisiae, as authors seem to claim: “In S. cerevisiae, uncharged tRNA binding to Gcn2 is essential for the response to AAS as well as to hydrogen peroxide and UV radiation, and these responses require Gcn1 (9)”

I suggest to substitute reference #16 (Murguía and Serrano, 2012), because it is a review, for the original one demonstrating an additional substrate for GCN2 kinase: Kwon NH, Kang T, Lee JY, Kim HH, Kim HR, Hong J, Oh YS, Han JM, Ku MJ, Lee SY, Kim S. Dual role of methionyl-tRNA synthetase in the regulation of translation and tumor suppressor activity of aminoacyl-tRNA synthetase-interacting multifunctional protein-3. Proc Natl Acad Sci U S A. 2011:108(49):19635-40. doi: 10.1073/pnas.1103922108.

Results

Page 6 (bottom): “…and showed a statistically significant decrease in body weight after E17.5 (Fig 2C)”, I guess figure referred is 2B instead of 2C.

Page 7 (end of first paragraph): “…and the gene expression levels of alveolar markers were comparable to those of the WT embryos between E18.5 and E19.5 (Fig 2E and F)”. Even though this sentence is not relevant for the global conclusion of the work, it does not reflect what it can be observed in the referred figure. According to the graphs in Fig. 2F, there is a significant difference in Aqp5 and Sp-c mRNA expression levels at E18.5 and E19.5.

Pages 7 (bottom) and 8 (top): most of the referred results regarding subcellular localization appear in the supplementary figure 2, but they have enough interest to be part of a main figure. Nevertheless, I would suggest to use MEF instead of HeLa cells and compare between WT and Gcn1l1∆RWDBD cells, given that subcellular localization of the protein could be relevant for its function.

Page 8 (middle paragraph): the phospho-specific western blot and the RT-PCR experiments strongly support the authors’ statement, but they would also consider to directly measure the subsequent increase in ATF4 translation upon eIF2α phosphorylation, by western blot.

Page 8 (bottom paragraph): results shown in figure 4A do not support authors’ statement, at least after 4 h of UV treatment. While GCN2 phosphorylation does not increase in WT cells and it is absent in mutant cells, it seems like eIF2α phosphorylation increases in both cell types. A reference to repeats or statistics is not present in the figure legend.

Page 9 (top paragraph): results regarding differences in cell growth and BrU incorporation seem to be clear, but the argued differences in cell size and in cell cycle seem to be not very prominent, even though statistically significant in some cases. Another question that could be apparently contradictory is the fact that with lower BrU incorporation mutant cells showed increased proportion in S and G2/M cell cycle phases.

Page 10, lines 7, 8, 9: in line 7 it should be “Fig. 6C and D” instead of Fig. 6C and 5D); in line 8 it should be “Fig. 6C” instead of “Fig. 6A”; in line 9 it should be “Fig. 6A” instead of “Fig. 6C”

Discussion

Page 11, line 6 it should be “Fig. 1 and 2” instead of “Fig. 1 and B”

Page 11, end of the first paragraph: authors state that heterozygous GCN1L1 KO mice express half amount of protein compared with wild type, but they do not experimentally demonstrate this point.

Page 13, first paragraph: authors argue that similarities between OLA1 and GCN1L1 KO mice and cells could be due to the reduction in ternary complex levels induced by both proteins in normal cells. This observation suggests that GCN1L1 function, although GCN2-independent, could still be related with eIF2α phosphorylation (reduced in mutant cells), raising the following question: is it possible that GCN1L1 could also regulate the activity of any of the other three eIF2α kinases (PKR, PERK, HRI)? Then, maybe authors should check this possibility by analysing the effect of the lack of GCN1L1 in the activation of those eIF2α kinases.

Reviewer #2: Yeast GCN1 is a multi-domain protein that associates with the RWD portion of GCN2 protein kinase, contributing to its activation and phosphorylation of eIF2 in response to amino acid depletion. Although GCN2 in mammalian cells has been well documented to induce the Integrated stress response in response to nutrient stresses, the role of the mammalian GCN1 ortholog (GCN1L1) has only sparsely studied. This manuscript addresses the function of GCN1L1 using mouse knockout mouse and in culture using MEF cells. The key findings of the manuscript are that loss of GCN1L1 function leads to a mouse growth defect that appears to involve delayed lung function. This growth defect is suggested to be independent of GCN2 as no growth defect has been reported for GCN2-deleted mice. Furthermore, this manuscript shows that mice expressing only GCN1l1 devoid of its GCN1-binding domain (RWD-BD) do not show the growth defect. Using MEF cells derived from this GCN1-modified mice, the manuscript shows that both GCN1 mutations thwart GCN2 activation in response to different stresses. There is also a cell cycle defect with loss of GCN1L1 that is appears to be independent of GCN2. Overall, these studies largely support the model presented in figure 7.

These results indicate that mammalian GCN1L1 not only functions in the activation of GCN2 (similar to yeast) but also has additional functions developmental and proliferative functions that are likely independent of GCN2. These findings are significant to the field and of broad interest. The manuscript is clearly written (although it should include more paragraph breaks) and experiments and appropriately designed and interpreted. There is a lot of enthusiasm for this manuscript, with only minor concerns.

Concerns:

1. Statistical analyses appear to be appropriate although there could be additional information (t-test one or two tailed?; Annova one or two way? Number of biological replicates?).

2. MW marker designation should be included in immunoblots. Quantitation with associated statistics should be included for key immunoblot measurements.

3. The last sentence in the results subsection titled “GCN1L1 was also necessary for the GCN2-mediated response to UV stress” describes Gcn1l1 KO MEFs for Fig 4A and 4B (should only be 4A . The figure 4 legend should also be clearer regarding panels A and B.

4. The proposed mechanism for the observed p21 increase in Gcn1l1ΔRWDBD and Gcn2 KO MEF cells presented in the discussion section appears to have an error. The second to last paragraph of the discussion section begins by proposing that p21 protein levels are increased by a decrease in ternary complex. However, ternary complex levels would be increased rather than decreased upon loss of eIF2 phosphorylation. This proposed mechanism should more clearly explain the observed increase in p21 levels that result from Gcn1l1ΔRWDBD or Gcn2 KO.

5. Deletion of the RWD binding domain of GCN1L1 in MEF cells reduced the total expression of GCN1L1 by greater that fifty percent (shown in Fig 3A) – therefore it is not entirely clear whether the resulting slowed growth, decreased Cdk1/cyclin B1, and increased p21 levels are a result of this decrease in expression or a result of the deletion of the RWD domain.

6. The decrease in cellular proliferation observed in Gcn1l1ΔRWDBD is not clearly attributed to either decreased Cdk1/cyclinB1 or increased p21. The manuscript cites both the increase in G2/M phase population (Fig. 5F) and the delay in entering S phase (Fig. 6E) of Gcn1l1ΔRWDBD MEFs as evidence for the decreased proliferation observed in Fig. 5A. Gcn1l1ΔRWDBD and Gcn2 KO both exhibit elevated p21 mRNA and protein levels, and while the Gcn1l1ΔRWDBD MEFs experienced a decrease in proliferation, the Gcn2 KO MEF cells did not. The delay in entering S phase observed in Fig 6E could be a result of increased p21 levels in Gcn1l1ΔRWDBD cells as compared to wild-type. Therefore, it may be necessary to address whether Gcn2 KO MEFs do not exhibit this delay in S phase entry, if in fact the delay in S phase contributes to reduced proliferation observed in Gcn1l1ΔRWDBD cells.

7. Perhaps it should be noted that humans with GCN2 defects have lung injuries.

Reviewer #3: There was some previous evidence that mammalian Gcn1 (mGcn1) is required for activation of mGcn2 based on overexpressing the IMPACT protein in mammalian cells, which competes for the RWD domain in mGcn2 for binding to mGcn1 and blunts the activation of mGcn2 and eIF2 phosphorylation in response to leucine starvation or UV treatment. Importantly, however, this paper provides direct evidence that Gcn1, and most likely its RWD binding domain (RWDBD), is required for Gcn2 activation in response to these stresses. They created two mGcn1 mutant mouse lines, one designed to knock out the entire protein and the other to eliminate only the RWDBD. Interestingly, these two mutant mice showed growth retardation and lethality, which was not observed for the Gcn2 KO mice, suggesting that mGcn1 has additional roles other than activation of Gcn2, and is essential for embryonic development. They also found reduced cell proliferation and G2/M arrest accompanying a decrease in Cdk1 and cyclinB1 in the mGcn1 ΔRWDBD MEFs, again not seen in mGcn2 KO MEFs, further suggesting a GCN2-independent role for mGcn1 in cell cycle regulation. Although the molecular mechanism involved in promoting expression of Cdk1 and cyclinB1 by mGcn1 remain to be determined, including which other RWD-containing protein is the relevant interaction partner for mGcn1 in the mechanism, the results are interesting and important nonetheless in uncovering this alternative function for mGcn1 in addition to providing strong, direct evidence that mGcn2 binding to the RWDBD of mGcn1 is required for Gcn2 activation by amino acid starvation and UV stress.

Major comments:

-p. 8. The statement: “Both the Asns and Psat1 genes were induced by depletion of methionine, serine or cystine in the WT MEFs, whereas the induction was largely suppressed in both Gcn1l1ΔRWDBD and Gcn2 KO MEFs (Fig 3D and E).” does not appear to be justified by the lack of a statistically significant reduction of Asns and Psat1 expression in the ΔRWDB mutant, and further evidence is required to claim that mGcn1 mediates the response to starvation for amino acids other than Leucine. One approach, which would enhance the story significantly would be to show that increased expression of Atf4, or an Atf4 reporter in response to different amino acid limitations in impaired in the ΔRWDBD MEFs.

-Based on the previous findings that In DRG2-knockdown HeLa cells, CDK1 and Cyclin B1 are decreased, while p21 is increased, the authors suggest in their model that Gcn1 functions in concert either DRG2 in stimulating Cdk1-cyclinB1 expression. If so, then knocking down DRG2 in the ΔRWDBD MEFs should not lead to any additional reduction in CDK1 and Cyclin B1 levels, but should do so in Gcn1+/+ MEFs. Performing this additional experiment has the potential of increasing the impact of this paper significantly by placing Gcn1 and Drg2 in the same pathway in regulating cell cycle, and as such, is strongly recommended.

Other comments:

-p.6, line 7 from bottom: In correct citation; should be Fig. 2B instead.

-p.7, lines 2-6: the legend to Fig. 2B lacks information that some of the data apparently came from experiments in which the gestational period was increased but other data did not; and this needs to be clarified.

-Fig. 2D-F: information is regarding the exact number of different animals of each genotype were examined and showed the indicated lung abnormalities or lung marker expression differences; and the statistical test applied in panel F.

-Fig. 3A lacks a control for equal protein loading.

-Fig. 3B lacks information about how many replicate experiments were conducted that gave a similar result.

-Fig. 3C-E lacks information about whether the replicates are biological replicates from independent RNA isolations or just technical replicates of the PCR amplifications of the same RNA samples.

-p.8 and Fig. 4B: Either replicate blots should be presented as supplementary material or (even better) quantification of blots from biological replicates should be provided for the P-eIF2a:eIF2a ratios to justify the conclusions that the ΔRWDBD mutant and the Gcn2 KO both reduce the proportion of eIF2a that is phosphorylated in response to UV.

-p. 8, 3rd line from bottom: cites Gcn1 KO incorrectly; should be Gcn2 KO according to Fig. 4B.

-Fig. 5C,D,F,and I: the number of replicates and the statistical tests employed should be indicated in the legend. Also, in panels C-D it is unclear if primary or immortalized MEFs were analyzed and should be stipulated in the legend.

-In Fig. S4B, it’s unclear if the amount of cleaved PARP and caspase-3 is increased relative to the uncleaved species, which also were higher in the mutant cells, indicating a need for quantifying the cleaved to full-length ratios from multiple replicates to justify the claim on p. 9. Also what would be the reason for seeing the increased cleavage only in the immortalized cells?

-the statement on p. 9: “Cell enlargement is often associated with cellular senescence (27); however, β-galactosidase staining showed similar intensities in both WT and Gcn1l1ΔRWDBD MEFs (data now shown)” needs to be backed up by showing the data and also explained in regard to whether or not the mutant cells show signs of senescence.

-p. 10, lines 2-5: It’s consistent with the data but not demonstrated by the results in Fig. 6A-B that Gcn1 is involved in the regulation of p21 expression by Gcn2.

-p. 10 and Figs. 6C,D,F: Either replicate blots should be presented as supplementary material or quantification of blots from biological replicates should be provided to justify the conclusions on p. 10 regarding the altered expression of Cdk1, cyclin B1, p21, and eIF2a proteins.

-p. 10: the statement: “Replenishment with fresh medium resulted in more WT MEFs in the S phase at 24 h, but at 28 h, fewer Gcn1l1ΔRWDBD MEFs and Gcn1l1ΔRWDBD MEFs were in the S phase compared to WT MEFs (Fig 6E).” has an obvious typo. In addition, it appears that there may be as many or more, not less, S-phase cells at 28h in the mutant vs. WT, and there are fewer mutant vs. WT cells in the G2/M phase at this time point. Also the data in Fig. 6E should be quantified from replicates as was done for Fig. 5F & I.

-It’s unclear why the data in Fig. 5E-F and 6E seem to give conflicting results about whether there is a delay at G2/M (Fig. 5) or in S (Fig. 6).

-Fig. 7A, the fact that the Gcn1-/- embryos die should be indicated

-p. 13: The sentence: “As both OLA1 and GCN1L1 can mediate stress-induced decreases in ternary complex formation, the phenotype observed for Gcn1l1ΔRWDBD may reflect the overlapping function of OLA1 and GCN1L1. As Gcn2 KO does not show a similar phenotype, such stresses, if any, are not likely conferred by AAS.” Doesn’t seem justified because the only known mechanism for Gcn1 to reduce ternary complex formation is via activation of Gcn2. Are they proposing that mGcn1 down-regulates ternary complex formation via DRG2? Is there evidence for this for the yeast proteins?

-p.13, regarding the statement: “As described above, p21 translation is increased by a decrease in the ternary complex. Indeed, the basal level of eIF2α phosphorylation appeared to decrease in both Gcn1l1ΔRWDBD and Gcn2 KO MEFs (Fig 4).” Unless I’m mistaken, the cited evidence showed that p21 translation is dependent on high-levels of the ternary complex; which seems at odds with the suggestion that increased p21 translation would result from decreased ternary complex formation owing to an elevated basal level of eIF2a phosphorylation in the ΔRWDBD mutant.

**Have all data underlying the figures and results presented in the manuscript been provided?**

Reviewer #1: Yes

Reviewer #2: Yes

Reviewer #3: No: Numerical data that underlies graphs or summary statistics should be provided in spreadsheet form as supporting information.

PLOS authors have the option to publish the peer review history of their article (what does this mean?). If published, this will include your full peer review and any attached files.

Reviewer #1: No

Reviewer #2: No

Reviewer #3: No

---

## [Editor Report · Decision Letter 1]

22 Feb 2020

Dear Dr Itoh,

We are pleased to inform you that your manuscript entitled "Ribosome binding protein GCN1 regulates the cell cycle and cell proliferation and is essential for the embryonic development of mice" has been editorially accepted for publication in PLOS Genetics. Congratulations!

Yours sincerely,

David Ron

Guest Editor

PLOS Genetics

Gregory Barsh

Editor-in-Chief

PLOS Genetics

Comments from the reviewers (if applicable):

**Data Deposition**

http://datadryad.org/submit?journalID=pgenetics&manu=PGENETICS-D-19-01400R1

**Press Queries**

---

## [Editor Report · Acceptance letter]

7 Apr 2020

PGENETICS-D-19-01400R1 

Ribosome binding protein GCN1 regulates the cell cycle and cell proliferation and is essential for the embryonic development of mice 

Dear Dr Itoh, 

We are pleased to inform you that your manuscript entitled "Ribosome binding protein GCN1 regulates the cell cycle and cell proliferation and is essential for the embryonic development of mice" has been formally accepted for publication in PLOS Genetics! Your manuscript is now with our production department and you will be notified of the publication date in due course.

With kind regards,

Matt Lyles

PLOS Genetics

On behalf of:
